

SciPost Phys. Lect. Notes 1 (2018)

# Minimal lectures on two-dimensional conformal field theory

**Sylvain Ribault**

Institut de Physique Théorique, Université Paris Saclay, CEA, F-91191 Gif-sur-Yvette, France

sylvain.ribault@ipht.fr

## Abstract

We provide a brief but self-contained review of conformal field theory on the Riemann sphere. We first introduce general axioms such as local conformal invariance, and derive Ward identities and BPZ equations. We then define minimal models and Liouville theory by specific axioms on their spectrums and degenerate fields. We solve these theories by computing three- and four-point functions, and discuss their existence and uniqueness.

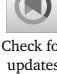
# 1   Introduction

Since the time of Euclid, mathematical objects are defined by axioms. Axiomatic definitions focus on the basic features of the defined objects, thereby avoiding alternative constructions that may be less fundamental. For example, in conformal field theory, the axiomatic approach (also called the conformal bootstrap approach) makes functional integrals unnecessary. We will define Liouville theory and minimal models by a sequence of axioms, starting with local conformal symmetry. Our axioms are necessary conditions. Their mutual consistency, in other words the existence of Liouville theory and minimal models, will be tested but not proved.

In the first three sections, most axioms are common to all two-dimensional conformal field theories. These axioms specify in particular how the Virasoro symmetry algebra acts on fields, and the existence and properties of the operator product expansion. Next, we introduce additional axioms that single out either Liouville theory, or minimal models. In particular, these axioms determine the three-point functions. Finally we check that these uniquely defined theories do exist, by studying their four-point functions. It is the success of such checks, much more than a priori considerations, that justifies our choices of axioms.

This text aims to be self-contained, except at the very end when we will refer to [1] for the properties of generic conformal blocks. For a more detailed text in the same spirit, see the review article [2]. For a wider and more advanced review, and a guide to the recent literature, see Teschner's text [3]. The Bible of rational conformal field theory is of course the epic textbook [4]. And Cardy's lecture notes [5] provide an introduction to the statistical physics applications of conformal field theory.

That these lectures are minimal does not just mean that they are relatively brief. This also means that they omit many concepts and assumptions that are usually introduced in two-dimensional conformal field theory, but that are not necessary for our purposes. Among these concepts and assumptions, let us mention the existence of a vacuum state, the notion of unitarity, the construction of fields as operators on the space of states, radial quantization, and consistency on Riemann surfaces other than the sphere. Minimalism is valuable not only for pedagogy, but also for research: we will follow shortcuts that may have been hard to see when originally solving Liouville theory and minimal models, but that can now be used when exploring other theories, for example non-diagonal theories [6].

# 2   The Virasoro algebra and its representations

## 2.1   Algebra

By definition, conformal transformations are transformations that preserve angles. In two dimensions with a complex coordinate $z$, any holomorphic transformation preserves angles. Infinitesimal conformal transformations are holomorphic functions close to the identity function,

$$z \mapsto z + \epsilon z^{n+1} \qquad (n \in \mathbb{Z}, \ \epsilon \ll 1) . \tag{2.1}$$

These transformations act on functions of $z$ via the differential operators

$$\ell_n = -z^{n+1}\frac{\partial}{\partial z} \, , \tag{2.2}$$

and these operators generate the Witt algebra, with commutation relations

$$[\ell_n, \ell_m] = (n-m)\ell_{m+n} \, . \tag{2.3}$$

The generators $(\ell_{-1}, \ell_0, \ell_1)$ generate an $s\ell_2$ subalgebra, called the algebra of infinitesimal global conformal transformations. The corresponding Lie group is the group of conformal transformations of the Riemann sphere $\mathbb{C} \cup \{\infty\}$,

$$z \mapsto \frac{az+b}{cz+d} \, . \tag{2.4}$$

**Exercise 2.1** (Global conformal group of the sphere)
*Show that the global conformal group of the sphere is $PSL_2(\mathbb{C})$, and includes translations, rotations, and dilatations.*

In a quantum theory, symmetry transformations act projectively on states. Projective representations of an algebra are equivalent to representations of a centrally extended algebra. This is why we always look for central extensions of symmetry algebras.

**Definition 2.2** (Virasoro algebra)
*The central extension of the Witt algebra is called the Virasoro algebra. It has the generators $(L_n)_{n \in \mathbb{Z}}$ and $\mathbf{1}$, and the commutation relations*

$$[\mathbf{1}, L_n] = 0 \quad , \quad [L_n, L_m] = (n-m)L_{n+m} + \frac{c}{12}(n-1)n(n+1)\delta_{n+m,0}\mathbf{1} \, , \tag{2.5}$$

*where the number $c$ is called the central charge. (The notation $c\mathbf{1}$ stands for a central generator that always has the same eigenvalue $c$ within a given conformal field theory.)*

**Exercise 2.3** (Uniqueness of the Virasoro algebra)
*Show that the Virasoro algebra is the unique central extension of the Witt algebra.*

## 2.2 Representations

The spectrum, i.e. the space of states, must be a representation of the Virasoro algebra. Let us now make assumptions on what type of representation it can be.

**Axiom 2.4** (Representations that can appear in the spectrum)
*The spectrum is a direct sum of irreducible representations. In the spectrum, $L_0$ is diagonalizable, and the real part of its eigenvalues is bounded from below.*

Why this special role for $L_0$? Because we want to interpret it as the energy operator. Since the corresponding Witt algebra generator $\ell_0$ generates dilatations, considering it as the energy operator amounts to consider the radial coordinate as the time. We however do not assume that $L_0$ eigenvalues are real or that the spectrum is a Hilbert space, as this would restrict the central charge to be real. The $L_0$ eigenvalue of an $L_0$ eigenvector is called its conformal dimension.

Let us consider an irreducible representation that is allowed by our axiom. There must be an $L_0$ eigenvector $|v\rangle$ with the smallest eigenvalue $\Delta$. Then $L_n|v\rangle$ is also an $L_0$ eigenvector,

$$L_0 L_n|v\rangle = L_n L_0|v\rangle + [L_0, L_n]|v\rangle = (\Delta - n)L_n|v\rangle \, . \tag{2.6}$$

If $n > 0$ we must have $L_n|v\rangle = 0$, and $|v\rangle$ is called a primary state.

**Definition 2.5** (Primary and descendent states, level, Verma module)
*A primary state with conformal dimension $\Delta$ is a state $|v\rangle$ such that*

$$L_0|v\rangle = \Delta|v\rangle \quad , \quad L_{n>0}|v\rangle = 0 . \tag{2.7}$$

*The Verma module $\mathcal{V}_\Delta$ is the representation whose basis is $\left\{\prod_{i=1}^k L_{-n_i}|v\rangle\right\}_{0<n_1\leq\cdots\leq n_k}$. The level of the state $\prod_{i=1}^k L_{-n_i}|v\rangle$ is $N = \sum_{i=1}^k n_i \geq 0$. A state of level $N \geq 1$ is called a descendent state.*

Let us plot a basis of primary and descendent states up to the level 3:

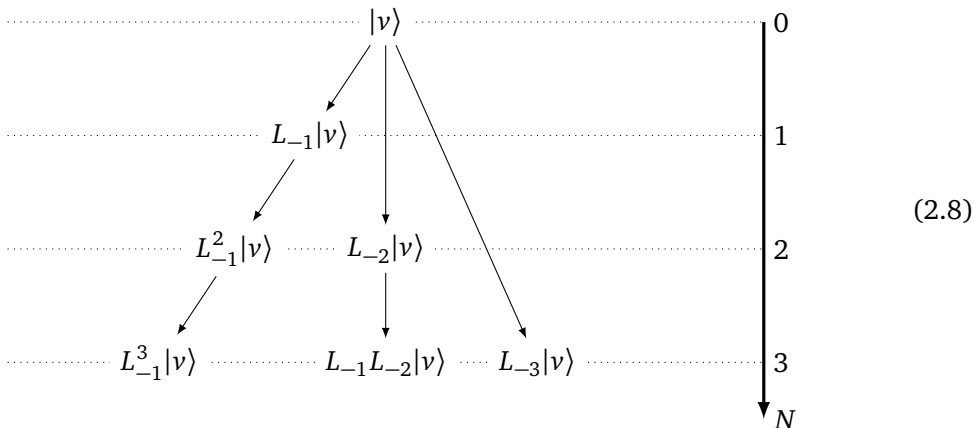

$$\tag{2.8}$$

We need not include the state $L_{-2}L_{-1}|v\rangle$, due to $L_{-2}L_{-1} = L_{-1}L_{-2} - L_{-3}$.

Are Verma module irreducible representations? i.e. do they have nontrivial subrepresentations? In any subrepresentation of a Verma module, $L_0$ is again diagonalizable and bounded from below, so there must be a primary state $|\chi\rangle$. If the subrepresentation differs from the Verma module, that primary state must differ from $|v\rangle$, and therefore be a descendent of $|v\rangle$.

## 2.3 Null vectors and degenerate representations

**Definition 2.6** (Null vectors)
*A descendent state that is also primary is called a null vector or singular vector.*

In the Verma module $\mathcal{V}_\Delta$, let us look for null vectors at the level $N = 1$. For $n \geq 1$ we have

$$L_n L_{-1}|v\rangle = [L_n, L_{-1}]|v\rangle = (n+1)L_{n-1}|v\rangle = \begin{cases} 0 & \text{if } n \geq 2 , \\ 2\Delta|v\rangle & \text{if } n = 1 . \end{cases} \tag{2.9}$$

So $L_{-1}|v\rangle$ is a null vector if and only if $\Delta = 0$, and the Verma module $\mathcal{V}_0$ is reducible. Let us now look for null vectors at the level $N = 2$. Let $|\chi\rangle = (L_{-1}^2 + aL_{-2})|v\rangle$, then $L_{n\geq 3}|\chi\rangle = 0$.

**Exercise 2.7**
*Compute $L_1|\chi\rangle$ and $L_2|\chi\rangle$, and find*

$$L_1|\chi\rangle = ((4\Delta + 2) + 3a)L_{-1}|v\rangle \quad , \quad L_2|\chi\rangle = \left(6\Delta + (4\Delta + \tfrac{1}{2}c)a\right)|v\rangle . \tag{2.10}$$

*Requiring that $L_1|\chi\rangle$ and $L_2|\chi\rangle$ vanish, find the coefficient a, and show that*

$$\Delta = \frac{1}{16}\left(5 - c \pm \sqrt{(c-1)(c-25)}\right) . \tag{2.11}$$

In order to simplify this formula, let us introduce other notations for $c$ and $\Delta$. We define

$$\text{the background charge } Q , \quad c = 1 + 6Q^2 , \quad \text{up to } Q \to -Q , \tag{2.12}$$

$$\text{the coupling constant } b , \quad Q = b + \frac{1}{b} , \quad \text{up to } b \to \pm b^{\pm 1} , \tag{2.13}$$

$$\text{the momentum } \alpha , \quad \Delta = \alpha(Q - \alpha) , \quad \text{up to reflections } \alpha \to Q - \alpha . \tag{2.14}$$

The condition (2.11) for the existence of a level two null vector becomes

$$\alpha = -\frac{1}{2} b^{\pm 1} . \tag{2.15}$$

To summarize, null vectors at levels 1 and 2 occur for particular values of $\Delta$. The null vectors at levels $N \leq 2$ can be written as $L_{\langle r,s \rangle}|v\rangle$ where $r,s$ are strictly positive integers such that $rs = N$,

| $N$ | $\langle r,s \rangle$ | $\Delta_{\langle r,s \rangle}$ | $\alpha_{\langle r,s \rangle}$ | $L_{\langle r,s \rangle}$ |
|---|---|---|---|---|
| 1 | $\langle 1,1 \rangle$ | $0$ | $0$ | $L_{-1}$ |
| 2 | $\langle 2,1 \rangle$ | $-\frac{1}{2} - \frac{3}{4}b^2$ | $-\frac{b}{2}$ | $L_{-1}^2 + b^2 L_{-2}$ |
|   | $\langle 1,2 \rangle$ | $-\frac{1}{2} - \frac{3}{4b^2}$ | $-\frac{1}{2b}$ | $L_{-1}^2 + b^{-2} L_{-2}$ |

$$\tag{2.16}$$

The pattern goes on at higher levels [4]: null vectors occur at level $N$ for finitely many dimensions $\Delta_{\langle r,s \rangle}$, with

$$\alpha_{\langle r,s \rangle} = \frac{Q}{2} - \frac{1}{2}(rb + sb^{-1}) . \tag{2.17}$$

(See Exercise 4.10 for a derivation.) If $\Delta \notin \{\Delta_{\langle r,s \rangle}\}_{r,s \in \mathbb{N}^*}$, then $\mathcal{V}_\Delta$ is irreducible. If $\Delta = \Delta_{\langle r,s \rangle}$, then $\mathcal{V}_\Delta$ contains a nontrivial submodule, generated by the null vector and its descendent states. For generic values of the central charge $c$, this submodule is the Verma module $\mathcal{V}_{\Delta_{\langle r,s \rangle}+rs}$.

**Definition 2.8** (Degenerate representation)
*The coset of the reducible Verma module $\mathcal{V}_{\Delta_{\langle r,s \rangle}}$, by its Verma submodule $\mathcal{V}_{\Delta_{\langle r,s \rangle}+rs}$, is an irreducible module $\mathcal{R}_{\langle r,s \rangle}$, that is called a degenerate representation:*

$$\mathcal{R}_{\langle r,s \rangle} = \frac{\mathcal{V}_{\Delta_{\langle r,s \rangle}}}{\mathcal{V}_{\Delta_{\langle r,s \rangle}+rs}} . \tag{2.18}$$

*In this representation, the null vector vanishes,*

$$L_{\langle r,s \rangle}|v\rangle = 0 . \tag{2.19}$$

The vanishing of null vectors will be crucial for solving Liouville theory and minimal models.

# 3 Conformal field theory

Now that we understand the algebraic structure of conformal symmetry in two dimensions, let us study how the Virasoro algebra acts on objects that live on the Riemann sphere – the fields of conformal field theory. (Technically, fields are sections of vector bundles over the sphere.)

### 3.1 Fields

**Axiom 3.1** (State-field correspondence)
*For any state $|w\rangle$ in the spectrum, there is an associated field $V_{|w\rangle}(z)$. The map $|w\rangle \mapsto V_{|w\rangle}(z)$ is linear and injective. We define the action of the Virasoro algebra on such fields as*

$$L_n V_{|w\rangle}(z) = L_n^{(z)} V_{|w\rangle}(z) = V_{L_n|w\rangle}(z) \, . \tag{3.1}$$

**Definition 3.2** (Primary field, descendent field, degenerate field)
*Let $|v\rangle$ be the primary state of the Verma module $\mathcal{V}_\Delta$. We define the primary field $V_\Delta(z) = V_{|v\rangle}(z)$. This field obeys*

$$L_{n\geq0} V_\Delta(z) = 0 \quad , \quad L_0 V_\Delta(z) = \Delta V_\Delta(z) \, . \tag{3.2}$$

*Similarly, descendent fields correspond to descendent states. And the degenerate field $V_{\langle r,s\rangle}(z)$ corresponds to the primary state of the degenerate representation $\mathcal{R}_{\langle r,s\rangle}$, and therefore obeys*

$$L_{\langle r,s\rangle} V_{\langle r,s\rangle}(z) = 0 \qquad \text{– for example,} \quad L_{-1} V_{\langle 1,1\rangle}(z) = 0 \, . \tag{3.3}$$

Now let us specify how fields depend on $z$.

**Axiom 3.3** (Dependence of fields on $z$)
*For any field $V(z)$, we have*

$$\frac{\partial}{\partial z} V(z) = L_{-1} V(z) \, . \tag{3.4}$$

Moreover, in this Section 3 we tacitly assume that all our fields are locally holomorphic i.e. $\frac{\partial}{\partial \bar{z}} V(z) = 0$. The dependence on $\bar{z}$ will be reintroduced and studied in Section 4.1.

Let us derive consequences of this axiom, starting with the $z$-dependence of the action $L_n^{(z)}$ of Virasoro generators on fields. On the one hand,

$$\frac{\partial}{\partial z}\left(L_n^{(z)} V(z)\right) = \left(\frac{\partial}{\partial z} L_n^{(z)}\right) V(z) + L_n^{(z)} \frac{\partial}{\partial z} V(z) \, . \tag{3.5}$$

On the other hand, using our axiom, we find

$$\frac{\partial}{\partial z}\left(L_n^{(z)} V(z)\right) = L_{-1}^{(z)} L_n^{(z)} V(z) = -(n+1) L_{n-1}^{(z)} V(z) + L_n^{(z)} L_{-1}^{(z)} V(z) \, . \tag{3.6}$$

This implies

$$\frac{\partial}{\partial z} L_n^{(z)} = -(n+1) L_{n-1}^{(z)} \, , \qquad (\forall n \in \mathbb{Z}) \, . \tag{3.7}$$

These infinitely many equations can be encoded into one functional equation,

$$\frac{\partial}{\partial z} \sum_{n\in\mathbb{Z}} \frac{L_n^{(z)}}{(y-z)^{n+2}} = 0 \, . \tag{3.8}$$

**Definition 3.4** (Energy-momentum tensor)
*The energy-momentum tensor is a field, that we define by the formal Laurent series*

$$T(y) = \sum_{n\in\mathbb{Z}} \frac{L_n^{(z)}}{(y-z)^{n+2}} \, . \tag{3.9}$$

*In other words, for any field $V(z)$, we have*

$$T(y)V(z) = \sum_{n\in\mathbb{Z}} \frac{L_n V(z)}{(y-z)^{n+2}} \quad , \quad L_n V(z) = \frac{1}{2\pi i} \oint_z dy \, (y-z)^{n+1} T(y) V(z) \, . \tag{3.10}$$

In the case of a primary field $V_\Delta(z)$, using eq. (3.4) and writing regular terms as $O(1)$, this definition reduces to

$$T(y)V_\Delta(z) = \frac{\Delta}{(y-z)^2}V_\Delta(z) + \frac{1}{y-z}\frac{\partial}{\partial z}V_\Delta(z) + O(1) \, . \tag{3.11}$$

This is our first example of an operator product expansion.

The energy-momentum tensor $T(y)$ is locally holomorphic as a function of $y$, and acquires poles in the presence of other fields. Since we are on the Riemann sphere, it must also be holomorphic at $y = \infty$.

**Axiom 3.5** (Behaviour of $T(y)$ at infinity)

$$T(y) \underset{y\to\infty}{=} O\left(\frac{1}{y^4}\right) \, . \tag{3.12}$$

To motivate this axiom, let us do some dimensional analysis. If $z$ has dimension $-1$, then according to eq. (3.4) $L_{-1}$ has dimension 1, and $T(y)$ has dimension 2. The dimensionless quantity that should be holomorphic at infinity is therefore the differential $T(y)dy^2$. At infinity a holomorphic coordinate is $\frac{1}{y}$ and a holomorphic differential is $d(\frac{1}{y}) = -\frac{dy}{y^2}$, so our axiom amounts to $T(y)dy^2 = O(\frac{dy^2}{y^4})$ being holomorphic.

## 3.2 Correlation functions and Ward identities

**Definition 3.6** (Correlation function)
*To $N$ fields $V_1(z_1),\ldots,V_N(z_N)$ with $i \neq j \implies z_i \neq z_j$, we associate a number called their correlation function or $N$-point function, and denoted as*

$$\left\langle V_1(z_1)\cdots V_N(z_N)\right\rangle \, . \tag{3.13}$$

*For example, $\left\langle \prod_{i=1}^N V_{\Delta_i}(z_i)\right\rangle$ is a function of $\{z_i\}, \{\Delta_i\}$ and $c$. Correlation functions depend linearly on fields, and in particular $\frac{\partial}{\partial z_1}\left\langle V_1(z_1)\cdots V_N(z_N)\right\rangle = \left\langle \frac{\partial}{\partial z_1}V_1(z_1)\cdots V_N(z_N)\right\rangle$.*

**Axiom 3.7** (Commutativity of fields)
*Correlation functions do not depend on the order of the fields,*

$$V_1(z_1)V_2(z_2) = V_2(z_2)V_1(z_1) \, . \tag{3.14}$$

Let us work out the consequences of conformal symmetry for correlation functions. In order to study an $N$-point function $Z$ of primary fields, we introduce an auxiliary $(N+1)$-point function $Z(y)$ where we insert the energy-momentum tensor,

$$Z = \left\langle \prod_{i=1}^N V_{\Delta_i}(z_i)\right\rangle \quad , \quad Z(y) = \left\langle T(y)\prod_{i=1}^N V_{\Delta_i}(z_i)\right\rangle \, . \tag{3.15}$$

$Z(y)$ is a meromorphic function of $y$, with poles at $y = z_i$, whose residues are given by eq. (3.11) (using the commutativity of fields). Moreover $T(y)$, and therefore also $Z(y)$, vanish in the limit $y \to \infty$. So $Z(y)$ is completely determined by its poles and residues,

$$Z(y) = \sum_{i=1}^N \left(\frac{\Delta_i}{(y-z_i)^2} + \frac{1}{y-z_i}\frac{\partial}{\partial z_i}\right)Z \, . \tag{3.16}$$

But $T(y)$ does not just vanish for $y \to \infty$, it behaves as $O(\frac{1}{y^4})$. So the coefficients of $y^{-1}, y^{-2}, y^{-3}$ in the large $y$ expansion of $Z(y)$ must vanish,

$$\sum_{i=1}^{N} \partial_{z_i} Z = \sum_{i=1}^{N} \left( z_i \partial_{z_i} + \Delta_i \right) Z = \sum_{i=1}^{N} \left( z_i^2 \partial_{z_i} + 2\Delta_i z_i \right) Z = 0 \ . \tag{3.17}$$

These three equations are called global Ward identities. They determine how $Z$ behaves under global conformal transformations of the Riemann sphere,

$$\left\langle \prod_{i=1}^{N} V_{\Delta_i} \left( \frac{az_i + b}{cz_i + d} \right) \right\rangle = \prod_{i=1}^{N} (cz_i + d)^{2\Delta_i} \left\langle \prod_{i=1}^{N} V_{\Delta_i}(z_i) \right\rangle \ . \tag{3.18}$$

Let us solve the global Ward identities in the cases of one, two, three and four-point functions. For a one-point function, we have

$$\partial_z \left\langle V_\Delta(z) \right\rangle = 0 \quad , \quad \Delta \left\langle V_\Delta(z) \right\rangle = 0 \ . \tag{3.19}$$

So one-point functions are constant, and non-vanishing only if $\Delta = 0$. Similarly, two-point functions must obey

$$\left\langle V_{\Delta_1}(z_1) V_{\Delta_2}(z_2) \right\rangle \propto (z_1 - z_2)^{-2\Delta_1} \quad , \quad (\Delta_1 - \Delta_2) \left\langle V_{\Delta_1}(z_1) V_{\Delta_2}(z_2) \right\rangle = 0 \ . \tag{3.20}$$

So a two-point function can be non-vanishing only if the two fields have the same dimension. For three-point functions, there are as many equations (3.17) as unknowns $z_1, z_2, z_3$, and therefore a unique solution with no constraints on $\Delta_i$,

$$\left\langle \prod_{i=1}^{3} V_{\Delta_i}(z_i) \right\rangle \propto (z_1 - z_2)^{\Delta_3 - \Delta_1 - \Delta_2} (z_1 - z_3)^{\Delta_2 - \Delta_1 - \Delta_3} (z_2 - z_3)^{\Delta_1 - \Delta_2 - \Delta_3} \ , \tag{3.21}$$

with an unknown proportionality coefficient that does not depend on $z_i$. For four-point functions, the general solution is

$$\left\langle \prod_{i=1}^{4} V_{\Delta_i}(z_i) \right\rangle = \left( \prod_{i<j} (z_i - z_j)^{\delta_{ij}} \right) G \left( \frac{(z_1 - z_2)(z_3 - z_4)}{(z_1 - z_3)(z_2 - z_4)} \right) \ , \tag{3.22}$$

where $G$ is an arbitrary function, and $\delta_{ij}$ are such that $\sum_{j \neq i} \delta_{ij} = -2\Delta_i$, where $\delta_{ij} \underset{i>j}{=} \delta_{ji}$. The six numbers $(\delta_{ij})_{i<j}$ are subject to only four equations, leaving two undetermined combinations. Changing these combinations amounts to a redefinition $G(z) \to z^\lambda (1-z)^\mu G(z)$.

So the three global Ward identities effectively reduce the four-point function to a function of one variable $G$ – equivalently, we can set $z_2, z_3, z_4$ to fixed values, and recover the four-point function from its dependence on $z_1$ alone.

**Exercise 3.8** (Conformal symmetry of four-point functions)
*Let us define $V_\Delta(\infty) = \lim_{z \to \infty} z^{2\Delta} V_\Delta(z)$. Check that this is finite when inserted into a two- or three-point function. More generally, show that this is finite using the behaviour (3.18) of correlation functions under $z \to -\frac{1}{z}$. Show that there is a (unique) choice of $\delta_{ij}$ such that*

$$G(z) = \left\langle V_{\Delta_1}(z) V_{\Delta_2}(0) V_{\Delta_3}(\infty) V_{\Delta_4}(1) \right\rangle \ . \tag{3.23}$$

We have been studying global conformal invariance of correlation functions of primary fields, rather than more general fields. This was not only for making things simpler, but also

because correlation functions of descendents can be deduced from correlation functions of primaries. For example,

$$\left\langle L_{-2}V_{\Delta_1}(z_1)V_{\Delta_2}(z_2)\cdots\right\rangle = \frac{1}{2\pi i}\oint_{z_1}\frac{dy}{y-z_1}Z(y) = \sum_{i=2}^{N}\left(\frac{1}{z_1-z_i}\frac{\partial}{\partial z_i}+\frac{\Delta_i}{(z_i-z_1)^2}\right)Z\ , \quad (3.24)$$

where we used first eq. (3.10) for $L_{-2}V_{\Delta_1}(z_1)$, and then eq. (3.16) for $Z(y)$. This can be generalized to any correlation function of descendent fields. The resulting equations are called local Ward identities.

## 3.3  Belavin–Polyakov–Zamolodchikov equations

Local and global Ward identities are all we can deduce from conformal symmetry. But correlation functions that involve degenerate fields obey additional equations.

For example, let us replace $V_{\Delta_1}(z_1)$ with the degenerate primary field $V_{\langle 1,1\rangle}(z_1)$ in our $N$-point function $Z$. Since $\frac{\partial}{\partial z_1}V_{\langle 1,1\rangle}(z_1) = L_{-1}V_{\langle 1,1\rangle}(z_1) = 0$, we obtain $\frac{\partial}{\partial z_1}Z = 0$. In the case $N = 3$, having $\Delta_1 = \Delta_{\langle 1,1\rangle} = 0$ in the three-point function (3.21) leads to

$$\left\langle V_{\langle 1,1\rangle}(z_1)V_{\Delta_2}(z_2)V_{\Delta_3}(z_3)\right\rangle \propto (z_1-z_2)^{\Delta_3-\Delta_2}(z_1-z_3)^{\Delta_2-\Delta_3}(z_2-z_3)^{-\Delta_2-\Delta_3}\ , \quad (3.25)$$

and further imposing $z_1$-independence leads to

$$\left\langle V_{\langle 1,1\rangle}(z_1)V_{\Delta_2}(z_2)V_{\Delta_3}(z_3)\right\rangle \neq 0 \quad\Longrightarrow\quad \Delta_2 = \Delta_3\ . \quad (3.26)$$

This coincides with the condition (3.20) that the two-point function $\left\langle V_{\Delta_2}(z_2)V_{\Delta_3}(z_3)\right\rangle$ does not vanish. Actually, the field $V_{\langle 1,1\rangle}$ is an identity field, i.e. a field whose presence does not affect correlation functions. (See Exercise 4.6.)

In the case of $V_{\langle 2,1\rangle}(z_1)$, we have

$$\left(L_{-1}^2 + b^2 L_{-2}\right)V_{\langle 2,1\rangle}(z_1) = 0 \qquad \text{so that} \qquad L_{-2}V_{\langle 2,1\rangle}(z_1) = -\frac{1}{b^2}\frac{\partial^2}{\partial z_1^2}V_{\langle 2,1\rangle}(z_1)\ . \quad (3.27)$$

Using the local Ward identity (3.24), this leads to the second-order Belavin–Polyakov–Zamolodchikov partial differential equation

$$\left(\frac{1}{b^2}\frac{\partial^2}{\partial z_1^2}+\sum_{i=2}^{N}\left(\frac{1}{z_1-z_i}\frac{\partial}{\partial z_i}+\frac{\Delta_i}{(z_1-z_i)^2}\right)\right)\left\langle V_{\langle 2,1\rangle}(z_1)\prod_{i=2}^{N}V_{\Delta_i}(z_i)\right\rangle = 0\ . \quad (3.28)$$

More generally, a correlation function with the degenerate field $V_{\langle r,s\rangle}$ obeys a partial differential equation of order $rs$.

**Exercise 3.9** (Second-order BPZ equation for a three-point function)
 *Show that*

$$\left\langle V_{\langle 2,1\rangle}V_{\Delta_2}V_{\Delta_3}\right\rangle \neq 0 \quad\Longrightarrow\quad 2(\Delta_2-\Delta_3)^2 + b^2(\Delta_2+\Delta_3) - 2\Delta_{\langle 2,1\rangle}^2 - b^2\Delta_{\langle 2,1\rangle} = 0\ . \quad (3.29)$$

*Up to reflections of momentums, show that this is equivalent to*

$$\alpha_2 = \alpha_3 \pm \frac{b}{2}\ . \quad (3.30)$$

In the case of a four-point function, the BPZ equation amounts to a differential equation for the function of one variable $G(z)$, which therefore belongs to a finite-dimensional space of solutions.

**Exercise 3.10** (BPZ second-order differential equation)

*Show that the second-order BPZ equation for* $G(z) = \left\langle V_{\langle 2,1\rangle}(z)V_{\Delta_1}(0)V_{\Delta_2}(\infty)V_{\Delta_3}(1)\right\rangle$ *is*

$$\left\{\frac{z(1-z)}{b^2}\frac{\partial^2}{\partial z^2} + (2z-1)\frac{\partial}{\partial z} + \Delta_{\langle 2,1\rangle} + \frac{\Delta_1}{z} - \Delta_2 + \frac{\Delta_3}{1-z}\right\} G(z) = 0\,, \qquad (3.31)$$

# 4 Conformal bootstrap

We have seen how conformal symmetry leads to linear equations for correlation functions: Ward identities and BPZ equations. In order to fully determine correlation functions, we need additional, nonlinear equations, and therefore additional axioms: single-valuedness of correlation functions, and existence of operator product expansions. Using these axioms for studying conformal field theories is called the conformal bootstrap method.

## 4.1 Single-valuedness

**Axiom 4.1** (Single-valuedness)

*Correlation functions are single-valued functions of the positions, i.e. they have trivial monodromies.*

Our two-point function (3.20) however has nontrivial monodromy unless $\Delta_1 \in \frac{1}{2}\mathbb{Z}$, as a result of solving holomorphic Ward identities. We would rather have a single-valued function of the type $|z_1 - z_2|^{-4\Delta_1} = (z_1 - z_2)^{-2\Delta_1}(\bar{z}_1 - \bar{z}_2)^{-2\Delta_1}$. This suggests that we need antiholomorphic Ward identities as well, and therefore a second copy of the Virasoro algebra.

**Axiom 4.2** (Left and right Virasoro algebras)

*We have two mutually commuting Virasoro symmetry algebras with the same central charge, called left-moving or holomorphic, and right-moving or antiholomorphic. Their generators are written $L_n, \bar{L}_n$, with in particular*

$$\frac{\partial}{\partial z}V(z) = L_{-1}V(z) \quad,\quad \frac{\partial}{\partial \bar{z}}V(z) = \bar{L}_{-1}V(z)\,. \qquad (4.1)$$

*The generators of conformal transformations are the diagonal combinations $L_n + \bar{L}_n$.*

Let us point out that we do not use the widespread notation $f(z,\bar{z})$ for a generic function, and $f(z)$ for a locally holomorphic function (i.e. a function such that $\frac{\partial}{\partial \bar{z}}f(z) = 0$). This notation comes from a complexification of the Riemann sphere that makes $z$ and $\bar{z}$ independent. For us, $\bar{z}$ is always the complex conjugate of $z$, so the notation $f(z,\bar{z})$ would be redundant.

Let us now consider left- and right-primary fields $V_{\Delta_i, \bar{\Delta}_i}(z_i)$. According to eq. (3.21), the three-point function of such fields is

$$\left\langle \prod_{i=1}^{3} V_{\Delta_i, \bar{\Delta}_i}(z_i)\right\rangle \propto (z_1 - z_2)^{\Delta_3 - \Delta_1 - \Delta_2}(\bar{z}_1 - \bar{z}_2)^{\bar{\Delta}_3 - \bar{\Delta}_1 - \bar{\Delta}_2} \times \cdots\,. \qquad (4.2)$$

Single-valuedness as a function of $z_i$ constrains the spins $s_i = \Delta_i - \bar{\Delta}_i$,

$$s_i + s_j - s_k \in \mathbb{Z}\,, \qquad (i \neq j \neq k)\,. \qquad (4.3)$$

This implies in particular $2s_i \in \mathbb{Z}$. In other words, any primary field $V_{\Delta, \bar{\Delta}}(z)$ must obey

$$\Delta - \bar{\Delta} \in \frac{1}{2}\mathbb{Z}\,. \qquad (4.4)$$

The simplest case is $\Delta = \bar{\Delta}$, which leads to the definition

**Definition 4.3** (Diagonal states, diagonal fields and diagonal spectrums)
*A primary state or field is called diagonal if it has the same left and right conformal dimensions. A spectrum is called diagonal if all primary states are diagonal.*

From now on we will use the notation $V_\Delta(z)$ for the diagonal field $V_{\Delta,\Delta}(z)$.

## 4.2 Operator product expansion and crossing symmetry

**Axiom 4.4** (Operator product expansion)
*Let $(|w_i\rangle)$ be a basis of the spectrum. There exist coefficients $C_{12}^i(z_1, z_2)$ such that we have the operator product expansion (OPE)*

$$V_{|w_1\rangle}(z_1) V_{|w_2\rangle}(z_2) = \sum_i C_{12}^i(z_1, z_2) V_{|w_i\rangle}(z_2) \,. \tag{4.5}$$

*In a correlation function, this sum converges for $z_1$ sufficiently close to $z_2$.*

While the linear equations of Section 3 relate $N$-point functions to other $N$-point function, OPEs allow us to reduce $N$-point functions to $(N-1)$-point functions. The price to pay for this reduction is the introduction of OPE coefficients. Using OPEs iteratively, we can actually reduce any correlation function to a combination of OPE coefficients, and two-point functions. (We stop at two-point functions because they are simple enough for being considered as known quantities.)

Let us study some properties of OPE coefficients. Assuming that the spectrum is made of diagonal primary states and their descendent states, the OPE of two primary fields is

$$V_{\Delta_1}(z_1) V_{\Delta_2}(z_2) = \sum_{\Delta \in S} C_{\Delta_1, \Delta_2, \Delta} |z_1 - z_2|^{2(\Delta - \Delta_1 - \Delta_2)} \Big( V_\Delta(z_2) + O(z_1 - z_2) \Big) \,, \tag{4.6}$$

where the subleading terms are contributions of descendents fields. In particular, the $z_1, z_2$-dependence of the coefficients is dictated by the behaviour (3.18) of correlation functions under translations $z_i \to z_i + c$ and dilatations $z_i \to \lambda z_i$, leaving a $z_i$-independent unknown factor $C_{\Delta_1, \Delta_2, \Delta}$. Then, as in correlation functions, contributions of descendents are deduced from contributions of primaries via local Ward identitites.

**Exercise 4.5** (Computing the OPE of primary fields)
*Compute the first subleading term in the OPE (4.6), and find*

$$O(z_1 - z_2) = \frac{\Delta + \Delta_1 - \Delta_2}{2\Delta} \Big( (z_1 - z_2) L_{-1} + (\bar{z}_1 - \bar{z}_2) \bar{L}_{-1} \Big) V_\Delta(z_2) + O((z_1 - z_2)^2) \,. \tag{4.7}$$

*Hints: Insert $\oint_C dz(z - z_2)^2 T(z)$ on both sides of the OPE, for a contour $C$ that encloses both $z_1$ and $z_2$. Compute the relevant contour integrals with the help of eq. (3.11).*

**Exercise 4.6** ($V_{\langle 1,1 \rangle}$ is an identity field)
*Using $\frac{\partial}{\partial z_1} V_{\langle 1,1 \rangle}(z_1) = 0$, show that the OPE of $V_{\langle 1,1 \rangle}$ with another primary field is of the form*

$$V_{\langle 1,1 \rangle}(z_1) V_\Delta(z_2) = C_\Delta V_\Delta(z_2) \,, \tag{4.8}$$

*where the subleading terms vanish. Inserting this OPE in a correlation function, show that the constant $C_\Delta$ actually does not depend on $\Delta$. Deduce that, up to a factor $C = C_\Delta$, the field $V_{\langle 1,1 \rangle}$ is an identity field.*

We will now express three- and four-point functions in terms of OPE coefficients, and deduce constraints on these coefficients. Inserting the OPE in a three-point function of primary fields, we find

$$\left\langle \prod_{i=1}^{3} V_{\Delta_i}(z_i) \right\rangle = \sum_{\Delta \in S} C_{\Delta_1,\Delta_2,\Delta} |z_1 - z_2|^{2(\Delta-\Delta_1-\Delta_2)} \left( \langle V_\Delta(z_2) V_{\Delta_3}(z_3) \rangle + O(z_1 - z_2) \right), \quad (4.9)$$

$$= C_{\Delta_1,\Delta_2,\Delta_3} |z_1 - z_2|^{2(\Delta_3-\Delta_1-\Delta_2)} \left( |z_2 - z_3|^{-4\Delta_3} + O(z_1 - z_2) \right), \quad (4.10)$$

assuming the two-point function is normalized as $\langle V_\Delta(z_1) V_\Delta(z_2) \rangle = |z_1 - z_2|^{-4\Delta}$. It follows that $C_{\Delta_1,\Delta_2,\Delta_3}$ coincides with the undertermined constant prefactor of the three-point function (4.2). This factor is called the three-point structure constant, and we have

$$\left\langle \prod_{i=1}^{3} V_{\Delta_i}(z_i) \right\rangle = C_{\Delta_1,\Delta_2,\Delta_3} |z_1 - z_2|^{2(\Delta_3-\Delta_1-\Delta_2)} |z_1 - z_3|^{2(\Delta_2-\Delta_1-\Delta_3)} |z_2 - z_3|^{2(\Delta_1-\Delta_2-\Delta_3)}. \quad (4.11)$$

Let us now insert the OPE in a four-point function of primary fields:

$$\left\langle V_{\Delta_1}(z) V_{\Delta_2}(0) V_{\Delta_3}(\infty) V_{\Delta_4}(1) \right\rangle = \sum_{\Delta \in S} C_{\Delta_1,\Delta_2,\Delta} |z|^{2(\Delta-\Delta_1-\Delta_2)}$$

$$\times \left( \left\langle V_\Delta(0) V_{\Delta_3}(\infty) V_{\Delta_4}(1) \right\rangle + O(z) \right), \quad (4.12)$$

$$= \sum_{\Delta \in S} C_{\Delta_1,\Delta_2,\Delta} C_{\Delta,\Delta_3,\Delta_4} |z|^{2(\Delta-\Delta_1-\Delta_2)} \left( 1 + O(z) \right). \quad (4.13)$$

The contributions of descendents factorize into those of left-moving descendents, generated by the operators $L_{n<0}$, and right-moving descendents, generated by $\bar{L}_{n<0}$. So the last factor has a holomorphic factorization such that

$$\left\langle V_{\Delta_1}(z) V_{\Delta_2}(0) V_{\Delta_3}(\infty) V_{\Delta_4}(1) \right\rangle = \sum_{\Delta \in S} C_{\Delta_1,\Delta_2,\Delta} C_{\Delta,\Delta_3,\Delta_4} \mathscr{F}_\Delta^{(s)}(z) \mathscr{F}_\Delta^{(s)}(\bar{z}). \quad (4.14)$$

**Definition 4.7** (Conformal block)
*The four-point conformal block on the sphere,*

$$\mathscr{F}_\Delta^{(s)}(z) = z^{\Delta-\Delta_1-\Delta_2} \left( 1 + O(z) \right), \quad (4.15)$$

*is the normalized contribution of the Verma module $\mathcal{V}_\Delta$ to a four-point function, obtained by summing over left-moving descendents. Its dependence on $c, \Delta_1, \Delta_2, \Delta_3, \Delta_4$ are kept implicit. The label $(s)$ stands for s-channel, we will soon see what this means.*

Conformal blocks are in principle known, as they are universal functions, entirely determined by conformal symmetry. This is analogous to characters of representations, also known as zero-point conformal blocks on the torus.

**Exercise 4.8** (Computing conformal blocks)
*Compute the conformal block $\mathscr{F}_\Delta^{(s)}(z)$ up to the order $O(z)$, and find*

$$\mathscr{F}_\Delta^{(s)}(z) = z^{\Delta-\Delta_1-\Delta_2} \left( 1 + \frac{(\Delta+\Delta_1-\Delta_2)(\Delta+\Delta_4-\Delta_3)}{2\Delta} z + O(z^2) \right). \quad (4.16)$$

*Show that the first-order term has a pole when the Verma module $\mathcal{V}_\Delta$ has a null vector at level one. Compute the residue of this pole. Compare the condition that this residue vanishes with the condition (3.26) that three-point functions involving $V_{\langle 1,1 \rangle}$ exist.*

Our axiom 3.7 on the commutativity of fields implies that the OPE is associative, and that we can use the OPE of any two fields in a four-point function. In particular, using the OPE of the first and fourth fields, we obtain

$$\left\langle V_{\Delta_1}(z)V_{\Delta_2}(0)V_{\Delta_3}(\infty)V_{\Delta_4}(1)\right\rangle = \sum_{\Delta\in S} C_{\Delta,\Delta_1,\Delta_4}C_{\Delta_2,\Delta_3,\Delta}\mathscr{F}^{(t)}_{\Delta}(z)\mathscr{F}^{(t)}_{\Delta}(\bar{z})\,, \tag{4.17}$$

where $\mathscr{F}^{(t)}_{\Delta}(z) = (z-1)^{\Delta-\Delta_1-\Delta_4}\left(1 + O(z-1)\right)$ is a $t$-channel conformal block. The equality of our two decompositions (4.14) and (4.17) of the four-point function is called crossing symmetry, schematically

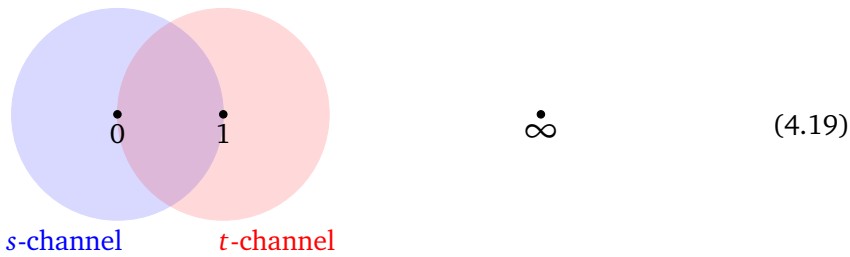

$$\sum_{\Delta_s\in S} C_{12s}C_{s34} \quad \underset{1}{\overset{2}{\diagup}}\!\!-\!\!s\!\!-\!\!\underset{4}{\overset{3}{\diagdown}} \quad = \sum_{\Delta_t\in S} C_{23t}C_{t41} \quad \underset{1}{\overset{2}{\bigvee}}\,t\,\underset{4}{\overset{3}{\bigvee}} \quad. \tag{4.18}$$

This equation holds if the sums on both sides converge, that is if $z$ is sufficiently close to both 0 and 1,

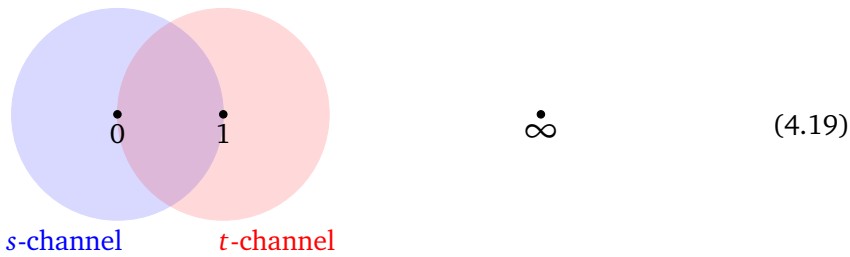

$s$-channel    $t$-channel

$$\overset{\bullet\!\!\bullet}{\infty} \tag{4.19}$$

Given the spectrum $S$, crossing symmetry is a system of quadratic equations for the structure constant $C_{\Delta_1,\Delta_2,\Delta_3}$. Requiring that this system has solutions is a strong constraint on the spectrum. In diagonal theories, crossing symmetry implies the existence of arbitrary correlation functions on the sphere [7], and not just of four-point functions.

## 4.3 Degenerate fields and the fusion product

Crossing symmetry equations are powerful, but typically involve infinite sums, which makes them difficult to solve. However, if at least one field is degenerate, then the four-point function belongs to the finite-dimensional space of solutions of a BPZ equation, and is therefore a combination of finitely many conformal blocks. For example, $\left\langle V_{\langle 2,1\rangle}(z)V_{\Delta_1}(0)V_{\Delta_2}(\infty)V_{\Delta_3}(1)\right\rangle$ is a combination of only two holomorphic $s$-channel conformal blocks. These two blocks are a particular basis of solutions of the BPZ equation (3.31). They are fully characterized by their asymptotic behaviour near $z=0$ (4.15), where the BPZ equation allows only two values of $\Delta$, namely $\Delta \in \{\Delta(\alpha_1 - \frac{b}{2}), \Delta(\alpha_1 + \frac{b}{2})\}$. The explicit expressions of these blocks are

$$\mathscr{F}^{(s)}_{\alpha_1-\frac{b}{2}}(z) = z^{b\alpha_1}(1-z)^{b\alpha_3}F(A,B,C,z) \quad , \quad \mathscr{F}^{(s)}_{\alpha_1+\frac{b}{2}}(z) = \mathscr{F}^{(s)}_{\alpha_1-\frac{b}{2}}(z)\Big|_{\alpha_1\to Q-\alpha_1}\,, \tag{4.20}$$

where $F(A,B,C,z)$ is the hypergeometric function with parameters

$$\begin{cases} A = \frac{1}{2} + b(\alpha_1+\alpha_3-Q) + b(\alpha_2-\frac{Q}{2})\,, \\ B = \frac{1}{2} + b(\alpha_1+\alpha_3-Q) - b(\alpha_2-\frac{Q}{2})\,, \\ C = 1 + b(2\alpha_1-Q)\,. \end{cases} \tag{4.21}$$

Another basis of solutions of the same BPZ equation is given by two $t$-channel blocks, which are characterized by their power-like behaviour near $z = 1$. Their explicit expressions are

$$\mathscr{F}^{(t)}_{\alpha_3-\frac{b}{2}}(z) = z^{b\alpha_1}(1-z)^{b\alpha_3}F(A,B,A+B-C+1,1-z) ,$$

$$\mathscr{F}^{(t)}_{\alpha_3+\frac{b}{2}}(z) = \mathscr{F}^{(t)}_{\alpha_3-\frac{b}{2}}(z)\Bigg|_{\alpha_3 \to Q-\alpha_3} . \tag{4.22}$$

The presence of only two $s$-channel fields with momentums $\alpha_1 \pm \frac{b}{2}$, and the constraint (3.30) on momentums in the three-point function $\left\langle V_{\langle 2,1\rangle}V_{\alpha_2}V_{\alpha_3}\right\rangle$, both mean that the operator product expansion $V_{\langle 2,1\rangle}(z)V_{\alpha_1}(0)$ involves only two primary fields $V_{\alpha_1\pm\frac{b}{2}}(0)$.

**Axiom 4.9** (Fusion product)
 *There is a bilinear, associative product of representations of the Virasoro algebra, that encodes the constraints on OPEs from Virasoro symmetry and null vectors. In particular,*

$$\mathscr{R}_{\langle 1,1\rangle} \times \mathscr{V}_\alpha = \mathscr{V}_\alpha \quad , \quad \mathscr{R}_{\langle 2,1\rangle} \times \mathscr{V}_\alpha = \sum_\pm \mathscr{V}_{\alpha\pm\frac{b}{2}} \quad , \quad \mathscr{R}_{\langle 1,2\rangle} \times \mathscr{V}_\alpha = \sum_\pm \mathscr{V}_{\alpha\pm\frac{1}{2b}} . \tag{4.23}$$

The fusion product can be defined algebraically [8]: the fusion product of two representations coincides with their tensor product as a vector space, where however the Virasoro algebra does not act as it would in the tensor product. (In the tensor product, central charges add.)
    Using the associativity of the fusion product, we have

$$\mathscr{R}_{\langle 2,1\rangle} \times \mathscr{R}_{\langle 2,1\rangle} \times \mathscr{V}_\alpha = \mathscr{R}_{\langle 2,1\rangle} \times \left(\sum_\pm \mathscr{V}_{\alpha\pm\frac{b}{2}}\right) = \mathscr{V}_{\alpha-b} + 2 \cdot \mathscr{V}_\alpha + \mathscr{V}_{\alpha+b} . \tag{4.24}$$

Since the fusion product of $\mathscr{R}_{\langle 2,1\rangle} \times \mathscr{R}_{\langle 2,1\rangle}$ with $\mathscr{V}_\alpha$ has finitely many terms, $\mathscr{R}_{\langle 2,1\rangle} \times \mathscr{R}_{\langle 2,1\rangle}$ must be a degenerate representation. On the other hand, eq. (4.23) implies that $\mathscr{R}_{\langle 2,1\rangle} \times \mathscr{R}_{\langle 2,1\rangle}$ is made of representations with momentums $\alpha_{\langle 2,1\rangle} \pm \frac{b}{2} = 0, -b$. The degenerate representation with momentum $0$ is $\mathscr{R}_{\langle 1,1\rangle}$. Calling $\mathscr{R}_{\langle 3,1\rangle}$ the degenerate representation with momentum $-b$, we just found

$$\mathscr{R}_{\langle 2,1\rangle} \times \mathscr{R}_{\langle 2,1\rangle} = \mathscr{R}_{\langle 1,1\rangle} + \mathscr{R}_{\langle 3,1\rangle} \quad , \quad \mathscr{R}_{\langle 3,1\rangle} \times \mathscr{V}_\alpha = \mathscr{V}_{\alpha-b} + \mathscr{V}_\alpha + \mathscr{V}_{\alpha+b} . \tag{4.25}$$

It can be checked that $\mathscr{R}_{\langle 3,1\rangle}$ has a vanishing null vector at level 3, so that our definition of $\mathscr{R}_{\langle 3,1\rangle}$ from fusion agrees with the definition from representation theory in Section 2.3.

**Exercise 4.10** (Higher degenerate representations)
 *Show that there exist degenerate representations $\mathscr{R}_{\langle r,s\rangle}$ (for $r,s \in \mathbb{N}^*$) with momentums $\alpha_{\langle r,s\rangle}$ (2.17), such that*

$$\mathscr{R}_{\langle r,s\rangle} \times \mathscr{V}_\alpha = \sum_{i=0}^{r-1}\sum_{j=0}^{s-1} \mathscr{V}_{\alpha+\alpha_{\langle r,s\rangle}+ib+jb^{-1}} , \tag{4.26}$$

$$\mathscr{R}_{\langle r_1,s_1\rangle} \times \mathscr{R}_{\langle r_2,s_2\rangle} = \sum_{r_3\overset{2}{=}|r_1-r_2|+1}^{r_1+r_2-1}\sum_{s_3\overset{2}{=}|s_1-s_2|+1}^{s_1+s_2-1} \mathscr{R}_{\langle r_3,s_3\rangle} , \tag{4.27}$$

*where the superscript in $\overset{2}{=}$ indicates that the corresponding sum runs by increments of* 2.

The knowledge of such fusion products will be crucial for defining and studying minimal models, whose spectrums are made of degenerate representations.

# 5 Minimal models and Liouville theory

Let us start the investigation of specific conformal field theories.

**Definition 5.1** (Conformal field theory)
*A (model of) conformal field theory on the Riemann sphere is a spectrum S and a set of correlation functions $\left\langle \prod_{i=1}^{N} V_{|w_i\rangle}(z_i) \right\rangle$ with $|w_i\rangle \in S$ that obey all our axioms, in particular crossing symmetry.*

**Definition 5.2** (Defining and solving)
*To define a conformal field theory is to give principles that uniquely determine its spectrum and correlation functions. To solve a conformal field theory is to actually compute them.*

In this Section we will define minimal models and Liouville theory. In Section 6 we will solve them.

## 5.1 Diagonal minimal models

**Definition 5.3** (Minimal model)
*A minimal model is a conformal field theory whose spectrum is made of finitely many irreducible representations of the product of the left and the right Virasoro algebras.*

Although there exist non-diagonal minimal models [4], we focus on diagonal minimal models, whose spectrums are of the type

$$S = \bigoplus_{\mathscr{R}} \mathscr{R} \otimes \bar{\mathscr{R}} \ , \tag{5.1}$$

where $\mathscr{R}$ and $\bar{\mathscr{R}}$ denote the same Virasoro representation, viewed as a representation of the left- or right-moving Virasoro algebra respectively.

**Axiom 5.4** (Degenerate spectrum)
*All representations that appear in the spectrum of a minimal model are degenerate.*

It is natural to use degenerate representations, because in an OPE of degenerate fields, only finitely many representations can appear. Conversely, we now assume that all representations that are allowed by fusion do appear in the spectrum, in other words

**Axiom 5.5** (Closure under fusion)
*The spectrum is closed under fusion.*

Let us assume that the spectrum contains a nontrivial degenerate representation such as $\mathscr{R}_{\langle 2,1 \rangle}$. Fusing it with itself, we get $\mathscr{R}_{\langle 1,1 \rangle}$ and $\mathscr{R}_{\langle 3,1 \rangle}$. Fusing multiple times, we get $(\mathscr{R}_{\langle r,1 \rangle})_{r \in \mathbb{N}^*}$ due to $\mathscr{R}_{\langle 2,1 \rangle} \times \mathscr{R}_{\langle r,1 \rangle} = \mathscr{R}_{\langle r-1,1 \rangle} + \mathscr{R}_{\langle r+1,1 \rangle}$. If the spectrum moreover contains $\mathscr{R}_{\langle 1,2 \rangle}$, then it must contain all degenerate representations.

**Definition 5.6** (Generalized minimal model)
*For any value of the central charge $c \in \mathbb{C}$, the generalized minimal model is the conformal field theory whose spectrum is*

$$S^{\text{GMM}} = \bigoplus_{r=1}^{\infty} \bigoplus_{s=1}^{\infty} \mathscr{R}_{\langle r,s \rangle} \otimes \bar{\mathscr{R}}_{\langle r,s \rangle} \ , \tag{5.2}$$

*assuming it exists and is unique.*

So, using only degenerate representations is not sufficient for building minimal models. In order to have even fewer fields in fusion products, let us consider representations that are multiply degenerate. For example, if $\mathscr{R}_{\langle 2,1 \rangle} = \mathscr{R}_{\langle 1,3 \rangle}$ has two vanishing null vectors, then $\mathscr{R}_{\langle 2,1 \rangle} \times \mathscr{R}_{\langle 2,1 \rangle} = \mathscr{R}_{\langle 1,1 \rangle}$ has only one term, as the term $\mathscr{R}_{\langle 3,1 \rangle}$ is not allowed by the fusion rules of $\mathscr{R}_{\langle 1,3 \rangle}$.

In order for a representation to have two null vectors, we however need a coincidence of the type $\Delta_{\langle r,s \rangle} = \Delta_{\langle r',s' \rangle}$. This is equivalent to $\alpha_{\langle r,s \rangle} \in \{\alpha_{\langle r',s' \rangle}, Q - \alpha_{\langle r',s' \rangle}\}$, and it follows that $b^2$ is rational,

$$b^2 = -\frac{q}{p} \qquad \text{with} \qquad \begin{cases} (p,q) \in \mathbb{N}^* \times \mathbb{Z}^* \\ p \wedge q = 1 \end{cases} \qquad \text{i.e.} \qquad c = 1 - 6\frac{(q-p)^2}{pq} \ . \tag{5.3}$$

For any integers $r, s$, we then have the coincidence

$$\Delta_{\langle r,s \rangle} = \Delta_{\langle p-r,q-s \rangle} \ . \tag{5.4}$$

In particular, for $1 \le r \le p-1$ and $1 \le s \le q-1$, there exists a doubly degenerate representation $\mathscr{R}_{\langle r,s \rangle} = \mathscr{R}_{\langle p-r,q-s \rangle}$. The diagonal spectrum built from representations of this type is

$$S_{p,q} = \frac{1}{2} \bigoplus_{r=1}^{p-1} \bigoplus_{s=1}^{q-1} \mathscr{R}_{\langle r,s \rangle} \otimes \bar{\mathscr{R}}_{\langle r,s \rangle} \ , \tag{5.5}$$

where the factor $\frac{1}{2}$ is here to avoid counting the same representation twice. This spectrum is not empty provided the coprime integers $p, q$ are both greater than 2,

$$p, q \ge 2 \ , \tag{5.6}$$

which implies in particular $b, Q \in i\mathbb{R}$ and $c < 1$. For other values of $p, q$, it turns out that no minimal models exist.

**Exercise 5.7** (Closure of minimal model spectrums under fusion)
*Show that $S_{p,q}$ is closed under fusion. If you are brave, compute the fusion products of the representations that appear in $S_{p,q}$. If you are very brave, show that $p, q \ge 2$ is a necessary condition for the existence of a finite, nontrivial set of multiply degenerate representations that closes under fusion.*

**Definition 5.8** (Diagonal minimal model)
*For $p, q \ge 2$ coprime integers, the $(p,q)$ minimal model is the conformal field theory whose spectrum is $S_{p,q}$, assuming it exists and is unique.*

For example, the minimal model with the central charge $c = \frac{1}{2}$ has the spectrum $S_{4,3}$,

$$\begin{cases} \Delta_{\langle 1,1 \rangle} = \Delta_{\langle 3,2 \rangle} = 0 \ , \\ \Delta_{\langle 1,2 \rangle} = \Delta_{\langle 3,1 \rangle} = \frac{1}{2} \ , \\ \Delta_{\langle 2,1 \rangle} = \Delta_{\langle 2,2 \rangle} = \frac{1}{16} \ . \end{cases} \iff \text{the Kac table} \quad \begin{array}{c|ccc} 2 & \frac{1}{2} & \frac{1}{16} & 0 \\ 1 & 0 & \frac{1}{16} & \frac{1}{2} \\ \hline & 1 & 2 & 3 \end{array} \tag{5.7}$$

## 5.2 Liouville theory

**Definition 5.9** (Liouville theory)
*For any value of the central charge $c \in \mathbb{C}$, Liouville theory is the conformal field theory whose spectrum is*

$$S^{\text{Liouville}} = \int_{\frac{Q}{2}+i\mathbb{R}_+} d\alpha \ \mathcal{V}_\alpha \otimes \bar{\mathcal{V}}_\alpha \ , \tag{5.8}$$

*and whose correlation functions are smooth functions of $b$ and $\alpha$, assuming it exists and its unique.*

Let us give some justification for this definition. We are looking for a diagonal theory whose spectrum is a continuum of representations of the Virasoro algebra. For $c \in \mathbb{R}$ it is natural to assume $\Delta \in \mathbb{R}$. Let us write this condition in terms of the momentum $\alpha$,

$$\Delta \in \mathbb{R} \iff \alpha \in \mathbb{R} \cup \left( \frac{Q}{2} + i\mathbb{R} \right) , \qquad \tag{5.9}$$

From Axiom 2.4, we need $\Delta$ to be bounded from below, and the natural bound is

$$\Delta_{\min} = \Delta\left( \alpha = \frac{Q}{2} \right) = \frac{Q^2}{4} = \frac{c-1}{24} . \tag{5.10}$$

This leads to $\alpha \in \frac{Q}{2} + i\mathbb{R}$. Assuming that each allowed representation appears only once in the spectrum, we actually restrict the momentums to $\alpha \in \frac{Q}{2} + i\mathbb{R}_+$, due to the reflection symmetry (2.14). We then obtain our guess (5.8) for the spectrum, equivalently $S^{\text{Liouville}} = \int_{\frac{c-1}{24}}^{\infty} d\Delta \; \mathcal{V}_\Delta \otimes \bar{\mathcal{V}}_\Delta$. We take this guess to hold not only for $c \in \mathbb{R}$, but also for $c \in \mathbb{C}$ by analyticity.

Other guesses for the lower bound may seem equally plausible, in particular $\Delta_{\min} = 0$. In the spirit of the axiomatic method, the arbiter for such guesses is the consistency of the resulting theory. This will be tested in Section 6.3, and the spectrum $S^{\text{Liouville}}$ will turn out to be correct.

Let us schematically write two- and three-point functions in Liouville theory, as well as OPEs:

$$\left\langle V_{\alpha_1} V_{\alpha_2} \right\rangle = B(\alpha_1) \delta(\alpha_1 - \alpha_2) , \tag{5.11}$$

$$\left\langle V_{\alpha_1} V_{\alpha_2} V_{\alpha_3} \right\rangle = C_{\alpha_1, \alpha_2, \alpha_3} , \tag{5.12}$$

$$V_{\alpha_1} V_{\alpha_2} = \int_{\frac{Q}{2} + i\mathbb{R}_+} d\alpha \; \frac{C_{\alpha_1, \alpha_2, \alpha}}{B(\alpha)} \left( V_\alpha + \cdots \right) , \tag{5.13}$$

where the expression for the OPE coefficient $\frac{C_{\alpha_1, \alpha_2, \alpha}}{B(\alpha)}$ in terms of two- and three-point structure constants is obtained by inserting the OPE into a three-point function. It would be possible to set $B(\alpha) = 1$ by renormalizing the primary fields $V_\alpha$, but this would prevent $C_{\alpha_1, \alpha_2, \alpha_3}$ from being a meromorphic function of the momentums, as we will see in Section 6.2.

In order to have reasonably simple crossing symmetry equations, we need degenerate fields. But the spectrum of Liouville theory is made of Verma modules, and does not involve any degenerate representations. In order to have degenerate fields, we need a special axiom:

**Axiom 5.10** (Degenerate fields in Liouville theory)
*The degenerate fields $V_{\langle r,s \rangle}$, and their correlation functions, exist.*

By the existence of degenerate fields, we also mean that such fields and their correlation functions obey suitable generalizations of our axioms. In particular, we generalize Axiom 4.4 by assuming that there exists an OPE between the degenerate field $V_{\langle 2,1 \rangle}$, and a field $V_\alpha$. However, according to the fusion rules (4.23), this OPE leads to fields with momentums $\alpha \pm \frac{b}{2}$, and in general $\alpha \in \frac{Q}{2} + i\mathbb{R} \implies (\alpha \pm \frac{b}{2}) \in \frac{Q}{2} + i\mathbb{R}$. We resort to the assumption in Definition 5.9 that correlation functions are smooth functions of $\alpha$, and take $V_\alpha$ to actually be defined for $\alpha \in \mathbb{C}$ by analytic continuation. (Or for $\alpha$ in a dense open subset of $\mathbb{C}$, if there are singularities.) This

allows us to write the OPE

$$V_{\langle 2,1 \rangle} V_\alpha \sim C_-(\alpha) V_{\alpha - \frac{b}{2}} + C_+(\alpha) V_{\alpha + \frac{b}{2}} \ , \tag{5.14}$$

where we introduced the degenerate OPE coefficients $C_\pm(\alpha)$.

# 6 Four-point functions

Let us determine the three-point structure constant by solving crossing symmetry equations. We begin with the equations that come from four-point functions with degenerate fields. These equations are enough for uniquely determining the three-point structure constant.

## 6.1 Single-valued four-point functions

The four-point function $G(z) = \left\langle V_{\langle 2,1 \rangle}(z) V_{\Delta_1}(0) V_{\Delta_2}(\infty) V_{\Delta_3}(1) \right\rangle$ obeys second-order BPZ equations in $z$ and $\bar{z}$. The most general solution of these two equations can be written in terms of the $s$-channel conformal blocks $\mathscr{F}^{(s)}_{\alpha_1 \pm \frac{b}{2}}$ of Section 4.3 as

$$G(z) = \sum_{i,j=\pm} c^{(s)}_{ij} \mathscr{F}^{(s)}_{\alpha_1 + i\frac{b}{2}}(z) \mathscr{F}^{(s)}_{\alpha_1 + j\frac{b}{2}}(\bar{z}) \ . \tag{6.1}$$

Let us determine how single-valuedness constrains the coefficients $c^{(s)}_{ij}$. Our conformal blocks have singularities at $z = 0, 1, \infty$, and single-valuedness is equivalent to $G(z)$ having trivial monodromy around $z = 0, 1$. Our $s$-channel decomposition is convenient for studying the monodromy around $z = 0$: near this point, each one of the two $s$-channel conformal blocks $\mathscr{F}^{(s)}_{\alpha_1 \pm \frac{b}{2}}(z)$ behaves as a power of $z$ (4.15), with its own exponent. For a generic value of $\alpha_1$, the difference of their two exponents is not integer, and $\mathscr{F}^{(s)}_{\alpha_1 - \frac{b}{2}}(z) \mathscr{F}^{(s)}_{\alpha_1 + \frac{b}{2}}(\bar{z})$ is not single-valued near $z = 0$. On the other hand, the two terms $\mathscr{F}^{(s)}_{\alpha_1 \pm \frac{b}{2}}(z) \mathscr{F}^{(s)}_{\alpha_1 \pm \frac{b}{2}}(\bar{z})$ are single-valued. We conclude that $c^{(s)}_{+-} = c^{(s)}_{-+} = 0$. Similarly, if we decompose the same four-point function in the $t$-channel basis of conformal blocks,

$$G(z) = \sum_{i,j=\pm} c^{(t)}_{ij} \mathscr{F}^{(t)}_{\alpha_1 + i\frac{b}{2}}(z) \mathscr{F}^{(t)}_{\alpha_1 + j\frac{b}{2}}(\bar{z}) \ , \tag{6.2}$$

then single-valuedness near $z = 1$ requires $c^{(t)}_{+-} = c^{(t)}_{-+} = 0$. Now the $s$- and $t$-channel bases of solutions of the BPZ equation must be related by a change of basis,

$$\mathscr{F}^{(s)}_{\alpha_1 + i\frac{b}{2}}(z) = \sum_{j=\pm} F_{ij} \mathscr{F}^{(t)}_{\alpha_1 + j\frac{b}{2}}(z) \ , \tag{6.3}$$

for some matrix $F_{ij}$ that is well-known in the case of our hypergeometric conformal blocks (4.20). Inserting this change of basis in our $s$-channel decomposition (6.1) of $G(z)$, we find a $t$-channel decomposition, with the coefficients

$$c^{(t)}_{i'j'} = \sum_{i,j=\pm} c^{(s)}_{ij} F_{ii'} F_{jj'} \ . \tag{6.4}$$

Inserting our single-valuedness conditions $c^{(t)}_{+-} = 0$ and $c^{(s)}_{-+} = c^{(s)}_{+-} = 0$ in this relation, we deduce

$$\frac{c^{(s)}_{++}}{c^{(s)}_{--}} = -\frac{F_{-+} F_{--}}{F_{++} F_{+-}} \ . \tag{6.5}$$

Using the known formula for $F_{ij}$, this is explicitly

$$\frac{c_{++}^{(s)}}{c_{--}^{(s)}} = \frac{\gamma(C)\gamma(C-1)}{\gamma(A)\gamma(B)\gamma(C-A)\gamma(C-B)} \quad \text{with} \quad \gamma(x) = \frac{\Gamma(x)}{\Gamma(1-x)}, \tag{6.6}$$

where the combinations $A, B, C$ of the momentums $\alpha_1, \alpha_2, \alpha_3$ are given in eq. (4.21).

This concludes the mathematical exercise of finding single-valued solutions of the hypergeometric equation. Now, in the case of Liouville theory, let us determine the coefficients $c_{ij}^{(s)}$ in terms of OPE coefficients and three-point structure constants. Using the degenerate OPE (5.14) and the three-point function (5.12), we find

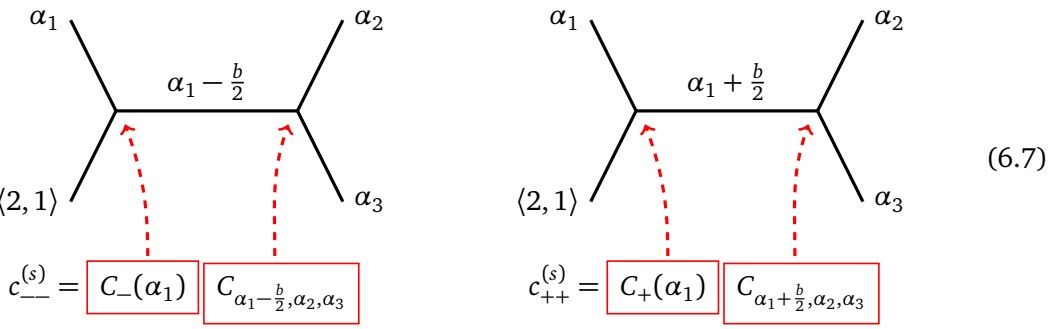

$$\tag{6.7}$$

Then eq. (6.6) becomes a shift equation for the dependence of $C_{\alpha_1,\alpha_2,\alpha_3}$ on $\alpha_1$,

$$\frac{C_+(\alpha_1)C_{\alpha_1+\frac{b}{2},\alpha_2,\alpha_3}}{C_-(\alpha_1)C_{\alpha_1-\frac{b}{2},\alpha_2,\alpha_3}} = \frac{\gamma(b(2\alpha_1-Q))\gamma(1+b(2\alpha_1-Q))}{\prod_{\pm,\pm}\gamma(\frac{1}{2}+b(\alpha_1-\frac{Q}{2})\pm b(\alpha_2-\frac{Q}{2})\pm b(\alpha_3-\frac{Q}{2}))}. \tag{6.8}$$

In order to find the three-point structure constant $C_{\alpha_1,\alpha_2,\alpha_3}$, we need to constrain the degenerate OPE coefficients $C_\pm(\alpha)$. To do this, we consider the special case where the last field is degenerate too, i.e. the four-point function $\langle V_{\langle 2,1\rangle}(z)V_\alpha(0)V_\alpha(\infty)V_{\langle 2,1\rangle}(1)\rangle$. In this case, using the degenerate OPE (5.14) twice, and the two-point function (5.11), we find

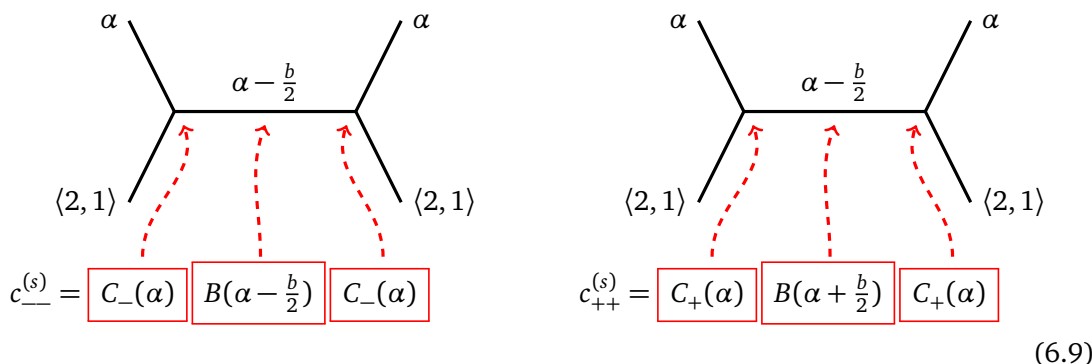

$$\tag{6.9}$$

Then eq. (6.6) boils down to

$$\frac{C_+(\alpha)^2 B(\alpha+\frac{b}{2})}{C_-(\alpha)^2 B(\alpha-\frac{b}{2})} = \frac{\gamma(b(2\alpha-Q))}{\gamma(-b(2\alpha-Q))}\frac{\gamma(-b^2-b(2\alpha-Q))}{\gamma(-b^2+b(2\alpha-Q))}. \tag{6.10}$$

Moreover, if we had the degenerate field $V_{\langle 1,2\rangle}$ instead of $V_{\langle 2,1\rangle}$ in our four-point functions, we would obtain the equations (6.8) and (6.10) with $b \to \frac{1}{b}$. Next, we will solve these equations.

## 6.2 Determining three-point structure constants

In order to solve the shift equations for $C_{\alpha_1,\alpha_2,\alpha_3}$, we need a function that produces Gamma functions when its argument is shifted by $b$ or $\frac{1}{b}$.

**Exercise 6.1** (Upsilon function)
*For $b > 0$, show that there is a unique (up to a constant factor) holomorphic function $\Upsilon_b(x)$ that obeys the shift equations*

$$\frac{\Upsilon_b(x+b)}{\Upsilon_b(x)} = b^{1-2bx}\gamma(bx) \qquad and \qquad \frac{\Upsilon_b(x+\frac{1}{b})}{\Upsilon_b(x)} = b^{\frac{2x}{b}-1}\gamma(\tfrac{x}{b}). \tag{6.11}$$

*For $ib > 0$, show that the meromorphic function*

$$\hat{\Upsilon}_b(x) = \frac{1}{\Upsilon_{ib}(-ix+ib)}, \tag{6.12}$$

*obeys shift equations that differ from eq. (6.11) by $b^{\cdots} \to (ib)^{\cdots}$.*

The function that solves the exercise can be written as

$$\Upsilon_b(x) = \lambda_b^{(\frac{Q}{2}-x)^2} \prod_{m,n=0}^{\infty} f\left(\frac{\frac{Q}{2}-x}{\frac{Q}{2}+mb+nb^{-1}}\right) \quad \text{with} \quad f(x) = (1-x^2)e^{x^2}, \tag{6.13}$$

where $\lambda_b$ is an unimportant $b$-dependent constant. The solution is unique for $b > 0$, because the ratio of two solutions would be a continuous function with aligned periods $b$ and $\frac{1}{b}$, and such a function must be constant if $b^2 \notin \mathbb{Q}$. In the complex plane, the periods $b$ and $\frac{1}{b}$ indeed look as follows:

$$
\begin{array}{cccc}
 & b > 0 & b \in \mathbb{C} & ib > 0 \\
 & c \geq 25 & c \in \mathbb{C} & c \leq 1
\end{array}
\tag{6.14}
$$

The formula (6.13) for $\Upsilon_b(x)$ actually makes sense not only for $b > 0$, but for $\Re b > 0$. This defines the functions $\Upsilon_b(x)$ and $\hat{\Upsilon}_b(x)$ for $\Re b > 0$ and $\Re ib > 0$ respectively, such that they are analytic in $b$.

Let us now solve the shift equation (6.8) using the function $\Upsilon_b$. We write the ansatz

$$C_{\alpha_1,\alpha_2,\alpha_3} = \frac{N_0 \prod_{i=1}^3 N(\alpha_i)}{\Upsilon_b(\alpha_1+\alpha_2+\alpha_3-Q)\Upsilon_b(\alpha_1+\alpha_2-\alpha_3)\Upsilon_b(\alpha_2+\alpha_3-\alpha_1)\Upsilon_b(\alpha_3+\alpha_1-\alpha_2)}, \tag{6.15}$$

where $N_0$ is a function of $b$, and $N(\alpha)$ is a function of $b$ and $\alpha$. The denominator of this ansatz takes care of the denominator of the shift equation, which therefore reduces to an equation that involves the dependence on $\alpha_1$ only,

$$\frac{C_-(\alpha_1)N(\alpha_1-\frac{b}{2})}{C_+(\alpha_1)N(\alpha_1+\frac{b}{2})} = \frac{b^{4b(2\alpha_1-Q)}}{\gamma(b(2\alpha_1-Q))\gamma(1+b(2\alpha_1-Q))}. \tag{6.16}$$

Combining this equation with the shift equation for $B(\alpha)$ (6.10), we can eliminate the unknown degenerate OPE coefficients $C_\pm(\alpha)$, and we obtain

$$\frac{(N^2B^{-1})(\alpha-\frac{b}{2})}{(N^2B^{-1})(\alpha+\frac{b}{2})} = b^{8b(2\alpha-Q)}\frac{\gamma(-b(2\alpha-Q))}{\gamma(b(2\alpha-Q))}\frac{\gamma(-b^2-b(2\alpha-Q))}{\gamma(-b^2+b(2\alpha-Q))}. \tag{6.17}$$

Together with its image under $b \to b^{-1}$, this equation has the solution

$$\left(N^2 B^{-1}\right)(\alpha) = \Upsilon_b(2\alpha)\Upsilon_b(2\alpha - Q) .$$ (6.18)

Therefore, we have only determined the combination $N^2 B^{-1}$, and not the individual functions $B$ and $N$ that appear in the two- and three-point functions. This is because we still have the freedom of performing changes of field normalization $V_\alpha(z) \to \lambda(\alpha)V_\alpha(z)$. Under such changes, we have $B \to \lambda^2 B$ and $N \to \lambda N$, while the combination $N^2 B^{-1}$ is invariant. Invariant quantities are the only ones that we can determine without choosing a normalization, and the only ones that will be needed for checking crossing symmetry.

It can nevertheless be convenient to choose a particular field normalization, when comparing different approaches to Liouville theory, or when considering specific applications. The normalization should then be chosen so that the structure constants have nice properties. For example, we should not choose the normalization such that $B(\alpha) = 1$, which would cause $N(\alpha)$ (and therefore $C_{\alpha_1, \alpha_2, \alpha_3}$) to have square-root branch cuts. We could rather adopt the field normalization such that $N(\alpha) = \Upsilon_b(2\alpha)$, which makes the three-point structure constant meromorphic as a function of the momentums. We also choose $N_0 = 1$, and we obtain the DOZZ formula (for Dorn, Otto, A. Zamolodchikov and Al. Zamolodchikov),

$$C_{\alpha_1, \alpha_2, \alpha_3} = \frac{\Upsilon_b(2\alpha_1)\Upsilon_b(2\alpha_2)\Upsilon_b(2\alpha_3)}{\Upsilon_b(\alpha_1 + \alpha_2 + \alpha_3 - Q)\Upsilon_b(\alpha_1 + \alpha_2 - \alpha_3)\Upsilon_b(\alpha_2 + \alpha_3 - \alpha_1)\Upsilon_b(\alpha_3 + \alpha_1 - \alpha_2)} .$$
(6.19)

For the original derivation of this formula from the Lagrangian formulation of Liouville theory, and the discussion of the formula in various limits, see [9].

The DOZZ formula holds if $c \notin ]-\infty, 1]$ i.e. $\Re b > 0$. On the other hand, doing the replacement $\Upsilon_b \to \hat{\Upsilon}_b$, we obtain a solution $\hat{C}$ that holds if $c \notin [25, \infty[$ i.e. $\Re ib > 0$, together with the corresponding functions $\hat{B}$ and $\hat{N}$. The solution of the shift equations is unique if $b$ and $b^{-1}$ are aligned, i.e. if $b^2 \in \mathbb{R}$. Therefore, for generic values of the central charge, both $C$ and $\hat{C}$ are solutions, and there are actually infinitely many other solutions. In order to prove the existence and uniqueness of Liouville theory, we will have to determine which solutions lead to crossing-symmetric four-point functions.

Our shift equations for three-point structure constants, and their solutions, are valid not only for Liouville theory, but also for (generalized) minimal models. In such models, the momentums $\alpha_{\langle r,s \rangle}$ belong to a lattice with periods $b$ and $\frac{1}{b}$, so they are uniquely determined by the shift equations. The solution is given by $C$ or $\hat{C}$, which coincide. (Actually $C$ has poles when $\alpha_i$ take degenerate values, one should take the residues.) This shows that (generalized) minimal models are unique, but not yet that they exist.

## 6.3 Crossing symmetry

We have found that (generalized) minimal models are unique, while Liouville theory is unique at least if $b^2 \in \mathbb{R}$. We will now address the question of their existence.

Using the $V_{\alpha_1} V_{\alpha_2}$ OPE (5.13), let us write the $s$-channel decomposition of a Liouville four-

point function,

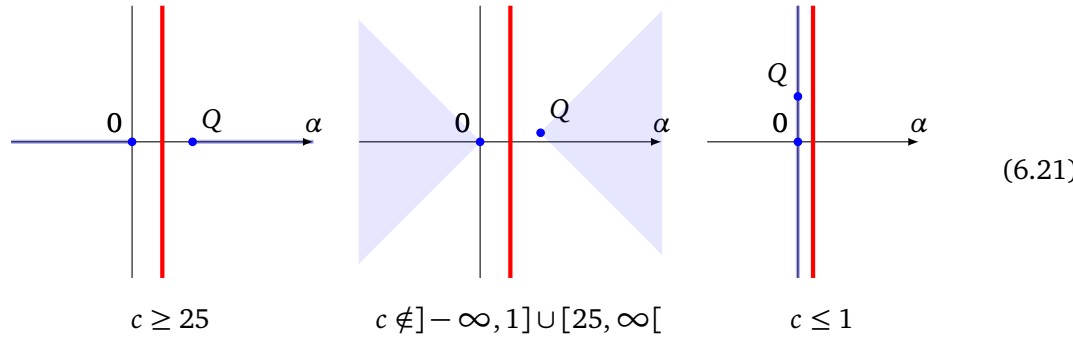

$$\left\langle V_{\alpha_1}(z) V_{\alpha_2}(0) V_{\alpha_3}(\infty) V_{\alpha_4}(1) \right\rangle = \int_{\frac{Q}{2}+i\mathbb{R}_+} d\alpha \; \boxed{\frac{C_{\alpha_1,\alpha_2,\alpha}}{B(\alpha)}} \; \boxed{C_{\alpha,\alpha_3,\alpha_4}} \; \mathscr{F}^{(s)}_\alpha(z) \mathscr{F}^{(s)}_\alpha(\bar{z})$$

(6.20)

(We have a similar expression with $B, C \to \hat{B}, \hat{C}$ whenever the solutions $\hat{B}, \hat{C}$ exist.) Let us accept for a moment that Liouville theory is crossing-symmetric if the central charge is such that the shift equations have unique solutions $C$ or $\hat{C}$, i.e. for $c \geq 25$ or $c \leq 1$ respectively. The integrand of our $s$-channel decomposition is well-defined, and analytic as a function of $b$, in the much larger regions $c \notin ]-\infty, 1]$ and $c \notin [25, \infty[$ respectively. If the integral itself was analytic as well, then crossing symmetry would hold in these regions by analyticity.

In order to investigate the analytic properties of the integral, let us first extend the integration half-line to a line, $\int_{\frac{Q}{2}+i\mathbb{R}_+} \to \frac{1}{2}\int_{\frac{Q}{2}+i\mathbb{R}}$. This is possible because the integrand is invariant under $\alpha \to Q - \alpha$: the conformal blocks are invariant because they are functions of the conformal dimension (2.14), and the combination $B(\alpha)^{-1} C_{\alpha_1,\alpha_2,\alpha} C_{\alpha,\alpha_3,\alpha_4}$ of structure constants is invariant as a consequence of $\Upsilon_b(\alpha) = \Upsilon_b(Q - \alpha)$. Let us then study the singularities of the integrand. We accept that the conformal blocks $\mathscr{F}^{(s)}_\alpha(z)$ have poles when $\alpha = \alpha_{\langle r,s \rangle}$ (2.17), the momentums for which the $s$-channel representation becomes reducible [1]. We now plot the positions of these poles (blue regions) relative to the integration line (red), depending on the central charge:

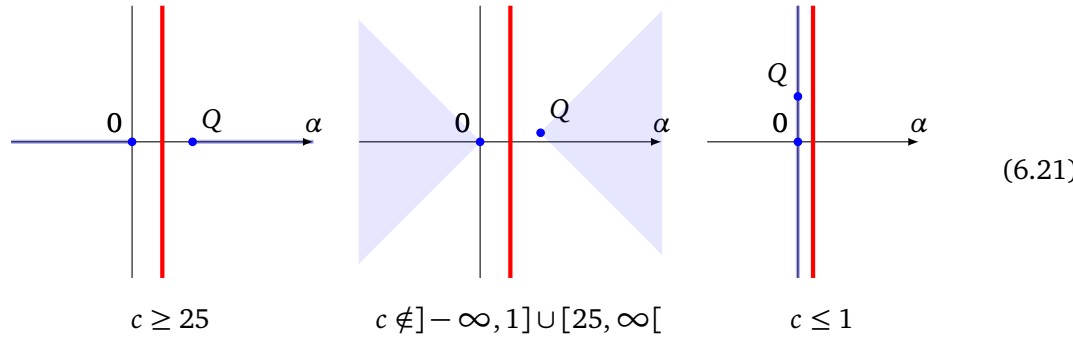

(6.21)

$c \geq 25$       $c \notin ]-\infty, 1] \cup [25, \infty[$       $c \leq 1$

When $c$ varies in the region $c \notin ]-\infty, 1]$, the poles never cross the integration line. Therefore, the four-point function built from $C$ is analytic on $c \notin ]-\infty, 1]$. So if Liouville theory exists for $c \geq 25$, then it also exists for $c \notin ]-\infty, 1]$, with the same structure constant $C$. On the other hand, if $c \leq 1$, then the poles are on the integration line, and actually the line has to be slightly shifted in order to avoid the poles. We cannot analytically continue the four-point function from the region $c \leq 1$ to complex values of $c$, because this would make infinitely many poles cross the integration line. So the structure constant $\hat{C}$ is expected to be valid for $c \leq 1$ only.

That is how far we can easily get with analytic considerations. Let us now seek input from numerical tests of crossing symmetry, using Al. Zamolodchikov's recursive formula for computing conformal blocks [1]. (See the associated Jupyter notebook, and the article [10].) We find that Liouville theory exists for all values of $c \in \mathbb{C}$, with the three-point structure constants $\hat{C}$ for $c \leq 1$, and $C$ otherwise. We also find that generalized minimal models exist

for all values of $c$, and minimal models exist at the discrete values (5.3) of $c \leq 1$ where they are defined. And we can numerically compute correlation functions with a good precision.

Historically, Liouville theory was first defined by quantizing a classical theory whose equation of motion is Liouville's equation. That definition actually gave its name to the theory. (That definition does not cover the case $c \leq 1$: using the name Liouville theory in this case, while natural in our approach, is not universally done at the time of this writing.) It can be shown that our definition of Liouville theory agrees with the historical definition, either by proving that the originally defined theory obeys our axioms, or by checking that both definitions lead to the same correlation functions, in particular the same three-point structure constants. See [3] for a guide to the literature on the construction of Liouville theory by quantization.

# Acknowledgements

I am grateful to the organizers of the Cargèse school, for challenging me to explain Liouville theory in about four hours. I with to thank Bertrand Eynard, Riccardo Guida and André Voros for helpful suggestions and comments. I am grateful to the participants of the Cargèse school, for their stimulating participation in the lectures. I wish to thank the SciPost editor and reviewers for their feedback and suggestions, which led to many improvements, both perturbative and non-perturbative.

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
