# Peer review of "Minimal lectures on two-dimensional conformal field theory"

_SciPost Physics Lecture Notes, doi:SciPost Phys. Lect. Notes 1 (2018)_

## Round 2 · Referee Report · Anonymous (Referee 1) · 2017-8-23

Strengths

  • A nice and clear introduction to the main CFT concepts and tools
  • The text can be read very easily in most parts.
  • It presents the important applications to minimal models and Liouville theory.

Weaknesses

  • The part devoted to the Liouville theory is a bit hard to follow.
  • There is a big jump in style from the previous sections to the ones devoted to Liouville
  • The list of references contains only one due to the author of the paper. It clearly needs to add a selection of references both elementary and advance.

Report

This paper is a brief introduction of conformal field theory (CFT)

that follows a "friendly" axiomatic approach to this vast subject.

It is in fact a recollection of Lectures delivered by the author at a Cargese School in 2016

on the conformal boostrap approach to Liouville theory.

The basic concepts and results are organized in definitions and axioms

that are clearly explained and illustrated with many examples.

This helps to understand step by step the connections underlying

CFT without pausing too much in each subject.

These lectures can be of great help to students trying to learn

the basic tools of CFT without going through the lengtly expositions available in the literature.

The author uses Liouville theory, and the minimal

models, as the main applications of CFT, which is certainly fine, but a mention

to other applications of CFT to Statistical Mechanics, Condensed Matter and String Theory

will be wellcome. The paper should contain a list of basic, and advanced, references

to guide the readers willing to expand their knowledge in CFT and Liouville theory.

There are already excellent articles and books that differ in the applications of CFT to physical models.

Requested changes

  • In section 2.1, the conformal fields are introduced in an algebraic manner using the Virasoro

algebra that was defined earlier in section 1.1. In particular, the energy-momentum tensor $T(z)$

arises naturally as a formal Laurent series of the position dependent Virasoro operators $L^{(z)}_n$.

In this axiomatic approach the OPE expansion given in eq.(2.11) seems to be a consequence of the

previous definitions and not a new axiom. Is that the case?

  • Axiom 2.7 states that the correlation functions do not depend on the order of the fields.

Here however, the fields are written only as functions of $z$, but not its conjugate $\bar{z}$.

The latter fact should be mentioned, and also the existence of two Virasoro algebras for the

holomorphic and antiholomorphic fields. This will be done in more detail in section 3.1

but a comment in this section is required.

  • A mention to contour deformation techniques is required to explain the derivation

of the Ward identity (2.16) using the OPE (2.11).

  • In Eq(2.20) left the exponent of $(z1-z2)$ must be $-2 $\Delta_1$ since it only involves the holomorphic part.

  • Eq.(2.6) reaches the same result as Eq.(2.20). It could be nice to make a comment.

  • The parametrization of the conformal blocks in eqs.(3.11), (3.12) and (3.13) does

not follow the standard radial ordering:

$$ <V(\infty) V(1) V(z) V(0)> (for |z| < 1) $$
or
$$ <V(\infty) V(z) V(1) V(0)> (for |z| > 1) $$
is there a reason for that? The above parametrization will help the reader

to understand more clearly the crossing symmetry Eq.(3.17) in combination of the associated plot in (3.18).

  • Definition (4.3) must be illustrated with Eqs.(4.5) and (4.7) that provide the

example of a minimal CFT. It should be perhaps said that when q = p+1, the CFT

is also unitary, being the first example the critical Ising model given in eq.(4.8).

A comment on the fact that the minimal c < 1 models the parameters Q and b are purely imaginary

will help later on to understand an important difference between these models and Liouville.

  • The definition of Generalized minimal model, should perhaps be postponed to the Liouville

section where they are considered in some detail.

-Section 4.2, that is devoted to the Liouville model, is a bit hard to follow because

it contains many technical details. First of all, it will help to recall Eq.(1.11) and notice the existence of imaginary

values of Delta when 1 < c < 25 that in sections 5.2 and 5.3 play an important role. The author

focus on technical details concerning complicated functions needed

to compute the three point structure constants Eq.(5.12) and to solve the crossing symmetry

relation. Probably the message contained on fig. (5.13) could be given before the technical

details are worked out. In my opinion this part have to be much improved.

---

## Round 2 · Referee Report · Anonymous (Referee 2) · 2017-9-6

Strengths

1) efficient introduction to conformal field theory in 2d. 2) useful reference

Weaknesses

1) later chapters are equation heavy and highly technical 2) the audience for the lecture notes is limited and maybe not entirely clear

Report

The author has given an abbreviated and somewhat dense introduction to conformal field theory in two dimensions.
The review appears to be a synopsis of his longer account ``Conformal field theory on the plane'' [1].
The review succeeds as lecture notes. A student would probably be better served by reading the original [1]. An expert
might find this review an efficient way of looking up certain results. A student who attended Ribault's Carg\`{e}se lectures
may find these notes a useful memory aid.
I have no objection to publication.

My personal experience with these lecture notes is that I understood Chapters 1 and 2 quite well, found Chapter 3 somewhat difficult to follow, and found Chapters 4 and 5 equation heavy and nearly opaque. In other words, I understood well the material I already knew, and understood quite poorly the chapters where my previous knowledge was weak.

If the author is serious about trying to improve the quality of the lecture notes, I would recommend lengthening to include more explanatory and background material, and also references to the literature. Of course I see the obvious danger is that the result will be a duplication of [1].
Perhaps the author can find a way to expand these lecture notes while at the same time maintaining a certain orthogonality to and independence from [1].

I leave the question of a major rewriting up to the author. The author should try to address my minimal set of comments below, however.

Requested changes

1) I did not understand the derivation of (2.12). It seems like the author derived instead the fact that $T(y) \sim (1/y^2)$ in the limit $y \to \infty$. Perhaps I do not understand his big O notation.

2) The presentation of the $\delta_{ij}$ below (2.22) was opaque. It seems there are four equations for six unknowns. Referring to [1] did not help clarify.

3) In exercise 2.8, I did not understand the expression for $V_\Delta(\infty)$. Is this a definition of what is meant by $V_\Delta(\infty)$? It might then help to change the notation since naively $V_\Delta(\infty)$ should be $\lim_{z \to \infty} V_\Delta(z)$ without extra factors of $z$. Also, I have a confusion about the power $z^{2 \Delta}$. Based on scaling, I would have guessed one should take $z^\Delta$ and not $z^{2 \Delta}$.

4) In Excercise 3.7, it would be useful to have a more precise question instead of a request to ``discuss''.

5) Given the subsequent ordering, it would make more sense to title section 4 ``Minimal models and Liouville theory".

---

## Round 2 · Referee Report · Anonymous (Referee 3) · 2017-9-27

Strengths

See report below.

Weaknesses

See report below.

Report

In the present form I can not recommend this review for publication.
However, with some effort it could be turned into a useful addition
to the literature.

My main criticism: In the current form the review creates an
impression to be essentially self-contained which it can not
fulfil. Important basic questions are left open, and no hint
whatsoever is given if this is simply the state of the art,
or if a better treatment can be found in the literature.

A first, fairly harmless, example occurs on page 15 before
eqn. (4.3). A sufficient condition is given for degeneracy
with no discussion why it is necessary.

A more serious one can be found on page 16. In order to
arrive at the crucial reflection relation (4.11) it is
argued that "It is however convenient to have fields...".
Most readers will wonder how the choice of the set of
fields in a theory can be a matter of convenience - the
usual bootstrap philosophy regards the set of fields that
can exist in a given theory as the solution to a rather
intricate problem. One may, of course, choose to add (4.11)
as an additional constraint to be imposed when solving the
bootstrap, but this raises the question how to motivate
this constraint. None of this is discussed in this review,
making me wonder how the treatment proposed in this review
can be useful for readers who do not use additional
sources of knowledge on the subject.

The only reference given in this review is a longer review
of the same author which does not offer much more
information on this and related points. The argument for the
reflection relation which can be found in the longer review
is not any better than the one in this paper,
appealing to the state-operator correspondence. In this
regard one should note that the author had only
introduced the postulate (here called "axiom") of
state-operator correspondence earlier in the text rather
than the stronger operator-state correspondence which is
often used in the axiomatics of CFT. This avoids partially
the issues with the more subtle nature of the operator-state
correspondence in Liouville theory. It allows fields not
belonging to the spectrum, which is importantly used in
the case of the degenerate fields. Any reader being
aware of this must wonder why one can not have two fields
with the same dimension, one belonging to the primary
state with the same dimension, the other not. None of
these issues is even mentioned in this review.

All this would not be too bad if the author would not
create the impression to give a self-contained treatment.
It would be much better to make the gaps in his arguments
visible, and to give pointers to the literature
where these issues and the motivations behind his "axioms"
have been discussed. The "minimalist" attitude concerning
references is part of the points severely limiting
the usefulness of this review.

To conclude: I think that this review needs a serious
revision to become useful. Even if brevity is the main
concern one can do better by adding just a few extra
pages. I'd strongly advocate to do so.

Requested changes

Substantial rewriting, see above.

---

## Round 3 · Referee Report · Anonymous (Referee 2) · 2017-12-22

Report

I am happy with the changes the author made to the manuscript. He has addressed my previous concerns.

---

## Round 3 · Referee Report · Anonymous (Referee 1) · 2017-12-22

Strengths

This paper is a brief introduction of conformal field theory (CFT)

that follows a friendly axiomatic approach to this vast subject.

Weaknesses

The section devoted to Liouvillel theory could be difficult to read to a non expert but
has been improved.

Report

The author has answered the questions posed in my previous report.
In particular the Liouville has been improved adding several figures to
explain the structures of the conformal blocks. Also a more complete
list of references have been included.

Requested changes

Please correct a typo after eq.(3.15)

for for s-channel -> for s-channel

---

## Round 3 · Referee Report · Anonymous (Referee 3) · 2018-1-9

Report

The suggestions made in my previous report have been taken into account only incompletely. Further changes are necessary to make the paper publishable.

  1. The most important remaining problem is the following. The set of axioms the author proposes to define the Liouville theory is not consistent, the crucial OPE (4.13) of degenerate fields contradicts the combination of Axiom 3.4 with Definition 4.9 (the spectrum of Liouville theory).

  2. It should also be noted that what is called Liouville theory in the paper under review is indeed equivalent to what is usually called quantum Liouville theory in most of the literature, defined as the quantum field theory obtained from the quantisation of a scalar field with dynamics given by Liouville's equation. This should be pointed out explicitly. If the author does not want to go into much detail about this equivalence, it would be necessary to at least offer some pointers to the literature concerning the evidence for this equivalence.

Two smaller points to be addressed are:

a) The argument on the top of page 18 that the lower bound for the spectrum of highest weights is (c-1)/24 is not compelling, without further arguments one could find the bound zero equally natural.

b) The statement "Invariant quantities are the only ones ... that matter physically" is quite confusing. The three point function is not invariant in this sense, yet usually considered to be physically relevant. It can, in particular, be related to expectation values of certain observables, which one certainly considers as physically meaningful quantities.

It seems that addressing especially the first of the points above requires a more substantial revision.

---

## Round 3 · Author Response

I am grateful to the reviewers and editor for their work. I have made many changes, especially in Sections 4 and 5, mainly in order to add or improve explanations. As a result, the revised version is more than 3 pages longer than the submitted version, while keeping the same plan. Let me comment on some of the most important changes, before addressing the reviewers’ specific concerns:

  1. In Sections 4 and 5, I have made the technical simplification of eliminating the reflection relation and the associated reflection coefficient. Instead, I have introduced a two-point structure constant $B(\alpha)$ in eq. (4.10). I hope that this makes the bootstrap analysis of Section 5 simpler, both conceptually and technically.

  2. I have added 8 references, including some general references that are cited in the Introduction. These general references can serve as guides to the original literature, a role that the present notes do not strive to fulfill. Most references are textbooks or review articles. They do not detract from the self-contained nature of these notes, up to one exception which is pointed out in the Introduction.

---

## Round 3 · List of Changes

Reply to Reviewer 1:

  1. Eq. (2.11) is indeed a specialization of eq. (2.10), as is now stated explicitly.

  2. The absence of antiholomorphic dependence in Section 2 is now stated explicitly after Axiom 2.3. Moreover, the role of $\bar z$ is now explained in more detail after Axiom 3.2.

  3. Actually, contour deformations are not needed for deriving (2.16). We only need to know the poles and residues of $Z(y)$.

  4. The mistaken exponent in (2.20) is now corrected.

  5. I have added a comment after (2.26) on its relation with (2.20), plus the Exercise 3.6 on the same subject.

  6. The reason for this choice of ordering can be understood graphically in eq. (3.18): it allows the $s$- and $t$-channels to correspond to the limits $z\to 0$ and $z\to 1$ respectively. On the other hand, radial ordering and radial quantization play no role in this text.

  7. Section 4.1 has been partly rewritten, I hope this answers some of the concerns about minimal models. In particular, there is now a comment on the values of $b, Q$ and $c$ after eq. (4.6). On the other hand I have refrained from commenting on unitarity, which plays no role in classifying and solving minimal models and Liouville theory. This concept would be superfluous in a minimal approach, and is already overemphasized in the existing literature, including in some of the cited references.

  8. In my terminology, generalized minimal models are models that exist for all values of $c$, and reduce to minimal models when $c$ takes appropriate discrete values. Therefore, generalized minimal models have much to do with minimal models, and little to do with Liouville theory, as is obvious from their spectrums.

  9. The values of $\Delta$ are real for states in the spectrum if $1<c<25$. The conformal dimensions of degenerate representations do become complex for these values of $c$, but this plays no role in the argument. I have followed the suggestion to give the figure (5.14) earlier in the text, and rewritten much of Section 5.

Reply to Reviewer 2:

I have done a major revision and expansion of Sections 4 and 5 in the hope of clarifying them, while also improving the rest of the text in a more perturbative fashion. In particular I have added explanations at the beginning of Section 3.2. Let me now address the ’minimal set of comments’:

  1. I have added more details in the derivation of (2.12).

  2. I now state that indeed there is an ambiguity in the choice of $\delta_{ij}$, and that this corresponds to different possible definitions of the function $G(z)$.

  3. In Exercise 2.8, I have given more justification of the definition of $V_\Delta(\infty)$. This notation is standard and unambiguous, so I am reluctant to change it.

  4. In what is now Exercise 3.8, I have given more precise guidance and questions.

  5. Right. I have reordered the title of Section 4.

Reply to Reviewer 3:

I would argue that this text is indeed self-contained, up to a small exception that is now stated explicitly in the Introduction. This text provides unambiguous definitions and solutions of Liouville theory and minimal models, that should be understandable without reference to the literature.

Admittedly there is relatively little discussion of the choices of axioms, but in the spirit of the axiomatico-deductive methods, axioms are justified a posteriori by the results that can be deduced from them (as I now recall in the Introduction). Admittedly there is also little discussion of the ’state of the art’: this discussion is now delegated to ref. [3].

Let me address some specific concerns:

  1. The condition for degeneracy in Section 4.1 has been rewritten. The condition (4.3) in the old version was actually not necessary. Doubly degenerate fields do not exist in minimal models only: they can exist whenever $b^2\in\mathbb{Q}$. This is now stated more explicitly, see also Exercise 4.7.

  2. The reflection relation has been eliminated. Moreover, in Section 4.2 I now explicitly assume that “each allowed representation appears only once in the spectrum”.

  3. Yes a conformal dimension can be shared by two primary states, one of them in the spectrum and the other not. This actually happens in Liouville theory with $c\leq 1$. This issue plays no role in solving the theory, so I would rather not discuss it in the text.

  4. I would be happy to plug any gaps in the argument, but I do not see any, so long I am allowed to choose my axioms. There are certainly gaps in referring to concepts and assumptions that are present in much of the literature, but not needed in the present argument. (See the Introduction.) Again, I hope that ref. [3] will help the curious reader.

---

## Round 4 · Author Response

List of changes
Reply to Reviewer 1: The typo after (3.15) is now corrected.
Reply to Reviewer 3:
-
There was indeed a contradition because Axiom 3.4 was too strong. I have modified that Axiom so that it applies only to fields that correspond to states in the spectrum. Then OPEs of degenerate fields are not covered, and should be discussed separately. I have added such a discussion after Axiom 4.10.
-
I have added a paragraph at the very end of Section 5, in order to make contact with the original definition (and naming) of Liouville theory, and to refer the reader to ref. [3] for more details.
-
I have added a paragraph between eqs. (4.10) and (4.11) in order to further discuss the guess for the lower bound, and to mention the alternative guess that the lower bound could be zero.
-
Between eqs. (5.18) and (5.19), I have amended the discussion of invariance under field renormalization: it is now stated that invariant quantities are sufficient for checking crossing symmetry, and explained why we may want to choose a normalization.
Additional changes:
-
At the end of the introduction, I have added a further motivation for the minimalistic approach, namely that it can be useful in research on other CFTs. Reference [6] is given as an example.
-
In Exercise 3.5, I have added substantial hints.

---

## Round 4 · List of Changes

Reply to Reviewer 1: The typo after (3.15) is now corrected.
Reply to Reviewer 3:
-
There was indeed a contradition because Axiom 3.4 was too strong. I have modified that Axiom so that it applies only to fields that correspond to states in the spectrum. Then OPEs of degenerate fields are not covered, and should be discussed separately. I have added such a discussion after Axiom 4.10.
-
I have added a paragraph at the very end of Section 5, in order to make contact with the original definition (and naming) of Liouville theory, and to refer the reader to ref. [3] for more details.
-
I have added a paragraph between eqs. (4.10) and (4.11) in order to further discuss the guess for the lower bound, and to mention the alternative guess that the lower bound could be zero.
-
Between eqs. (5.18) and (5.19), I have amended the discussion of invariance under field renormalization: it is now stated that invariant quantities are sufficient for checking crossing symmetry, and explained why we may want to choose a normalization.
Additional changes:
-
At the end of the introduction, I have added a further motivation for the minimalistic approach, namely that it can be useful in research on other CFTs. Reference [6] is given as an example.
-
In Exercise 3.5, I have added substantial hints.

---

## Editorial Decision

published